# How can instructions and feedback with external focus be shaped to enhance motor learning in children? A systematic review

**Ingrid P. A. van der Veer**[1]*, **Evi Verbecque**[1], **Eugene A. A. Rameckers**[1,2,3], **Caroline H. G. Bastiaenen**[4‡], **Katrijn Klingels**[1‡]

**1** Faculty of Rehabilitation Sciences and Physiotherapy, Rehabilitation Research Centre—REVAL, Hasselt University, Hasselt, Belgium, **2** Department of Functioning and Rehabilitation, Research School CAPHRI, Maastricht University, Maastricht, The Netherlands, **3** Centre of Expertise, Adelante Rehabilitation Centre, Valkenburg, The Netherlands, **4** Department of Epidemiology, Functioning, Participation & Rehabilitation, Research School CAPHRI, Maastricht University, Maastricht, The Netherlands

‡ These authors are joint senior authors on this work
* Ingrid.vanderveer@uhasselt.be

**Data Availability Statement:** All relevant data are within the manuscript and its Supporting Information files.

## Abstract

### Aim

This systematic review investigates the effectiveness of instructions and feedback with external focus applied with reduced frequency, self-controlled timing and/or in visual or auditory form, on the performance of functional gross motor tasks in children aged 2 to 18 with typical or atypical development.

### Methods

Four databases (PubMed, Web of Science, Scopus, Embase) were systematically searched (last updated May 31st 2021). Inclusion criteria were: 1. children aged 2 to 18 years old; 2. Instructions/feedback with external focus applied with reduced frequency, self-controlled timing, and/or visual or auditory form as intervention, to learn functional gross motor tasks; 3. Instructions/feedback with external focus applied with continuous frequency, instructor-controlled timing, and/or verbal form as control; 4. performance measure as outcome; 5. (randomized) controlled studies. Article selection and risk of bias assessment (with the Cochrane risk of bias tools) was conducted by two reviewers independently. Due to heterogeneity in study characteristics and incompleteness of the reported data, a best-evidence synthesis was performed.

### Results

Thirteen studies of low methodological quality were included, investigating effectiveness of reduced frequencies (n = 8), self-controlled timing (n = 5) and visual form (n = 1) on motor performance of inexperienced typically (n = 348) and atypically (n = 195) developing children, for acquisition, retention and/or transfer. For accuracy, conflicting or no evidence was found for most comparisons, at most time points. However, there was moderate evidence that self-controlled feedback was most effective for retention, and limited evidence that

**Funding:** The author(s) received no specific funding for this work.

**Competing interests:** The authors have declared that no competing interests exist.

visual analogy was most effective for retention and transfer. To improve quality of movement, there was limited evidence that continuous frequency was most effective for retention and transfer.

## Conclusion

More methodologically sound studies are needed to draw conclusions about the preferred frequency, timing or form. However, we cautiously advise considering self-controlled feedback, visual instructions, and continuous frequency.

## Trial registration

Registration: Prospero CRD42021225723. https://www.crd.york.ac.uk/prospero/display_record.php?ID=CRD42021225723.

## Introduction

Children apply many different gross motor skills in a wide variety of contexts, such as physical education (PE) classes, sports and playtime [1]. These so-called functional skills are defined as motor skills used in sports or other daily life activities that entail relatively complex movement organization [2]. Most children learn these skills almost effortlessly. Their increasing gross motor competence results from the interaction between factors in child (e.g. age, executive functions, psychological characteristics, and motor skill level), task (e.g. rules of the game, type of task, and level of task complexity) and environment (e.g. opportunities for PE and sports) [1,3–5]. However, motor skills learning can be challenging for some children, due to neurological conditions [6,7] or neurodevelopmental disorders [8–11]. Motor learning can be defined as a set of processes associated with practice or experience leading to relatively permanent improvements in the capability for producing motor skills [12]. Instructors, like PE teachers, trainers, coaches, and occupational and physical therapists, apply motor learning on a daily basis [13–16]. They use various motor learning variables, such as instructions and feedback, which they adapt to the child and the task practised [15–19]. Their instructions and feedback are shaped by parameters, such as content (e.g. a specific focus of attention), frequency, form (e.g. visual or verbal), and timing (self- or instructor-controlled) [18,20,21].

With implicit motor learning, a child learns without awareness and with no or minimal increase in verbal knowledge [22]. It is suggested that children benefit from this type of learning, because there is minimal involvement of the working memory [2,23,24]. Implicit motor learning can, for instance, be shaped by using an external focus of attention (EF) [23]. With an EF, the child's attention is directed to the impact of the movement on the environment [25]. On the contrary, with an internal focus of attention (IF) the attention is directed to its body movements [25]. According to the constrained action hypothesis, an IF promotes a larger involvement of cognitive processes due to a greater reliance on conscious control strategies. These strategies interfere with the normal automatic control processes of the motor system. An EF promotes these automatic control processes, therefore, enhancing motor learning more [26]. A recent systematic review investigated effectiveness of implicit learning strategies in functional motor skills learning in typically developing children (TDC) [23]. They concluded that the use of an EF appeared to be as, or even more, effective than an IF [23]. An EF was also more effective than an IF in motor learning for children with Mild Intellectual Disabilities

(MID) [27] and Attention Deficit and Hyperactivity Disorder (ADHD) [28]. However, an IF appeared more effective in children with Autism Spectrum Disorder (ASD) [29]. In children with Developmental Coordination Disorder (DCD), no differences were found for retention and transfer between groups using an EF or an IF [30,31]. Although, the beneficial effects of the EF have not yet been shown for each population, the constrained action hypothesis promotes using an EF for teaching motor skills [26]. Therefore, this systematic review focuses instructions and feedback with EF.

When using an EF in practical settings, instructors have to decide how often (frequency), when (timing) and in what form to provide their instructions and feedback (20). Feedback can be provided after each trial (continuous frequency) or after a number of trials (reduced frequency) [32–34]. Based on the guidance hypothesis, a reduced frequency would be more beneficial for retention and transfer than a continuous frequency because it reduces the feedback dependency enhancing the processing of other sources of information, which results in more implicit learning [34]. In stroke patients, it is indicated that reduced frequency is preferred [35]. However, in (a)typically developing children, this remains unclear [32,33]. The timing of instructions and feedback can be determined by the instructor (instructor-controlled) or the child (self-controlled) [36]. Self-controlled timing advances a child's autonomy, which is essential to enhance intrinsic motivation according to the Self-Determination Theory [37]. As motivation is considered relevant in motor learning, self-controlled timing could be more effective [38]. Studies in children showed that self-controlled feedback may enhance motor learning more than instructor-controlled feedback [36]. Most instructions and feedback are provided verbally [23,32,36] but instructors also use visual, tactile, and auditory (e.g. sound beeps) forms [14,17,19,20]. Currently, it remains unclear what frequency, form and timing are to be preferred when using instructions and feedback with EF [14,32,36].

While previous reviews suggest that the effectiveness of EF may be moderated by child and task characteristics, like working memory capacity, motor skill level and type of task [23,36], we hypothesize that the effectiveness of EF may also be moderated by the instructors' chosen frequency, timing, and form. Therefore, this systematic review investigates the effectiveness of instructions and feedback with EF applied with reduced frequency, in visual or auditory forms, and/or on request of the child (I), compared to instructions and feedback with EF applied with continuous frequency, in verbal form, and/or initiated by the instructor (C), on the performance of functional gross motor tasks (O) in children aged 2 to 18 with typical and atypical development (P).

## Methods

A systematic review of randomized controlled trials (RCTs) and non-randomized controlled clinical trials (CCTs) was performed. The hypotheses were: 1. instructions and feedback with EF applied with reduced frequency will be more effective than those applied with continuous frequency; 2. self-controlled instructions and feedback with EF will be more effective than instructor-controlled instructions and feedback; and 3. visual or auditory instructions and feedback with EF will be more effective than verbal instructions and feedback. This systematic review is written according to the Preferred Reporting Items for Systematic Reviews and Meta-analyses 2020 (PRISMA 2020) [39,40] and registered in the international prospective register of systematic reviews (PROSPERO) under registration number: CRD42021225723.

### Inclusion and exclusion criteria

Inclusion and exclusion criteria were defined in line with the PICOT structure (Population, Intervention, Control, Outcome, Type of study).

Inclusion criteria were:

1. Population: Children with (a)typical development aged 2–18 years. Studies which included a combined population of adolescents and adults were included if there were sub-analyses with adolescents.

2. Intervention: Instructions or feedback with EF applied with reduced frequency, in visual or auditory form and/or with self-controlled timing, used to learn functional gross motor tasks. With instructions or feedback with EF the instructor directs the attention of the child to the effects of the movement on the environment (e.g. "Try to focus on the red markers and try to keep the markers at the same height" when balancing a stabilometer) [25]. With Knowledge of Results feedback (KR) the instructor informs the child about the effects of the movement on the environment (e.g. by indicating to what extent the ball deviated the target in direction and distance) [41]. This information serves as a basis for error corrections improving next performances [34]. Although in KR the child needs to process the obtained information more to determine how to act, both EF and KR focus on the effects of the movement on the environment. Therefore, we considered KR as a subtype of feedback with EF. An analogy, a metaphor that integrates the complex structure of the to-be-learned task [42], is considered an EF because a child aims to reproduce the metaphor [38]. Reduced frequencies can be applied in fixed frequency (feedback after a fixed number of trials) or faded frequency (reducing the frequency over time) [32,35].

3. Control: Instructions and feedback with EF applied with continuous frequency, in verbal form and/or with instructor-controlled timing.

4. Outcome: A performance measure (e.g. accuracy or quality of movement) as primary outcome, used to assess acquisition and/or learning of functional gross motor tasks. Acquisition is measured during practice blocks or with a post-intervention test ("post-test"), and learning is measured with retention and/or transfer tests [43].

5. Type of study: Studies using a RCT or CCT without randomization design.

6. Publication type: Publications of original RCTs and CCTs.

7. Language: Studies written in English or Dutch.

Exclusion criteria were:

1. Population: Children with (a)typical development under the age of 2 years or adults.

2. Intervention: Instructions or feedback with an IF; intervention methods like Neuromotor Task Training, because they provide no insight into effectiveness of separate instructions or feedback; instructions and feedback used to learn laboratory, fine motor and static balance tasks, because they did not meet the definition of functional gross motor task [2].

3. Control: A tactile form of instructions and feedback, because it directs the attention of the child to the body, therefore, promoting an IF.

4. Outcome: Outcome measures that assessed brain anatomy and functions as primary outcomes.

5. Type of study: Studies performed with designs other than RCT and non-randomized CCT.

6. Publication type: Conference proceedings/reports and books.

7. Language: Studies not written in English or Dutch.

## Literature search

A systematic search was conducted in PubMed, Web of Science, Scopus and Embase. The search was last updated on the 31st of May 2021. Because instructions and feedback are also used when applying practice conditions, a broad search query was used to ensure that no relevant studies were missed. The search terms concerned four key topics: motor learning, instruction, feedback, and practice conditions. These topics were combined as motor learning AND (instruction OR feedback OR practice conditions). An explorative search to inventories relevant search terms showed that, in title and abstract, participants were often described in general (e.g. subjects). It also showed that various outcome measures were used to assess motor task performance (e.g. accuracy, speed, count, distance). To prevent studies being missed, search terms did not incorporate terms related to population or outcome. No date restrictions or filters were applied. See S1 File for the detailed search queries.

## Study selection

The eligibility of the studies was assessed in two phases: on title and abstract (phase 1); on full text (phase 2). The selection criteria were applied in a fixed sequence (population, intervention, control, outcome, type of study, publication type and language) by two reviewers independently (IvdV and EV). If necessary, authors were contacted for full texts. After each phase, a consensus meeting discussed the results of the article selection. Full text versions were read in case of disagreement after phase 1 and an independent reviewer (ER) was consulted in case of disagreement after phase 2. References of the included studies and of the three systematic reviews concerning children's motor learning (23,32,36) were checked by one reviewer (IvdV) to ensure that all relevant studies had been included.

## Data extraction

Data were extracted using a standardized sheet by one reviewer (IvdV or EV) and checked and complemented by the other. Corrections and additions were discussed between both reviewers; in the case of disagreement, an independent reviewer (ER) was consulted. Authors were not contacted for further details about studies.

For each study, the following data were extracted: 1. Characteristics of the study design: information regarding the group allocation of the participants (e.g. randomization procedure), blinding of participants, assessors, outcome measures and all relevant data for analyses; 2. Population characteristics: number of participants in total and per group, age range, mean age and standard deviations (SD), skill level (inexperienced or trained), and diagnosis, if given; 3. Intervention characteristics: details about instructions or feedback to the experimental and control group(s), the task, and the practice sessions (e.g. frequency, volume and duration); 4. Outcome and assessment time points: the primary and secondary outcome(s) to measure motor performance and type and timing of measurements in acquisition and test phase (pre-, post-, retention and/or transfer tests); 5. Results: summary statistics with measures of precision for each group, the data for differences between groups, and thresholds of minimal clinically important differences.

## Methodological quality assessment

The revised Risk of Bias tool (RoB2), for randomized trials [44], and the Risk of Bias in Non-randomized Studies of Interventions (ROBINS-I) [45], were used to assess methodological quality.

The RoB2 evaluates five major domains of biases: selection, performance, detection, attrition, and reporting biases. Signalling questions were answered to reach a domain-specific RoB judgement of 'low', 'some concerns' or 'high'. If not referred to a registered trial protocol, Questions 5.2 and 5.3 were answered based on the data-analysis section. Using the judgements of the five domains, an overall RoB judgement was made. If at least four domains were of some concern, the overall RoB was considered high.

The ROBINS-I evaluates seven major domains of biases: confounding, selection, classification, performance, detection, attrition, and reporting biases. As for the RoB2, signalling questions were used to reach a domain-specific RoB judgement of 'low', 'moderate', 'serious', 'critical' or 'no information'. If not referred to a registered trial protocol, Questions 7.1, 7.2 and 7.3 were answered based on the data-analysis section. Based on the domain-specific judgements, an overall RoB judgement was made.

Four reviewers (IvdV, EV, ER and KK) investigated RoB. Each study was assessed by two reviewers independently. A consensus meeting was organized with all reviewers and an epidemiologist (CB) to reach consensus.

## Analyses

Results were described for study selection, study characteristics and methodological quality. The RoB judgments were visualized [46]. To answer the hypotheses, as a first step a meta-analysis was planned with studies comparable for study design, instructions and feedback, and task. Therefore, the instructions and feedback were coded according to each parameter (frequency, timing and form). For frequency, the intervention was coded as reduced fixed or reduced faded frequency and the control as continuous frequency (hypothesis 1). For timing, the intervention was coded as self-controlled and the control as instructor-controlled (hypothesis 2). In studies investigating timing, the control group is either yoked (the children received feedback as their counterpart in the intervention group requested feedback) or instructor-controlled (the instructor determined when the child received feedback). Because of the chosen focus of this systematic review in the self-controlled aspect, we combined both yoked and instructor-controlled groups as control intervention. For form, the intervention was coded as visual or auditory and the control as verbal (hypothesis 3). Studies were grouped according to the type of comparison between coded intervention and control. Each task is defined by its own constraints, which are related to the context in which the task is performed [47]. Only studies with similar tasks could be combined in a meta-analysis. After subgrouping in subsequent steps according to (firstly) task and (secondly) population (TDC and per diagnosis), it was still not possible to pool data due to heterogeneity and to the incompleteness of the reported data. Therefore, a best-evidence synthesis was performed. The best-evidence synthesis table was structured according to the parameter of interest (frequency, timing, or form) and subdivided into comparisons of coded interventions and controls, as described above. If studies included more than one group with reduced frequency, the frequency that was most comparable with other studies was used for analysis. Within comparisons, studies were ordered according to comparable tasks and population, mentioning studies of good methodological quality first to increase the prominence of the most trustworthy evidence. This study aimed to investigate whether the instructor-controlled parameters frequency, timing and form moderate effectiveness of instructions and feedback in children. Subsequent analyses with sub groups were not performed for two reasons: 1. it was not possible to define relevant sub groups due to insufficient insights, and presented data in the included studies, into which child characteristics could be potentially relevant to moderate effectiveness [36]; and 2. the number of studies per potential comparison and methodological quality was too low. Results were

described per outcome measure. The results of each study were rated as significant (favouring a specific frequency, timing or form), inconsistent or not significant [48]. Then, the evidence for each comparison was rated according to the guidelines of van Tulder et al. [48]: strong (consistent findings among multiple high quality RCTs), moderate (consistent findings among multiple low quality RCTs and/or CCTs and/or one high quality RCT), limited (one low quality RCT and/or CCT), conflicting (inconsistent findings among multiple RCTs and/or CCTs), or no evidence from trials (no RCTs or CCTs). Consistency was defined as 75% of the studies assessing the same comparison showing results in the same direction.

## Results

### Study selection

The search resulted in 3813 unique hits. After screening title and abstract, 3521 hits were excluded. The reviewers agreed in 86% of the studies on inclusion or exclusion, 14% of the abstracts were discussed. The remaining 292 hits were screened on full text, eight of which met the inclusion criteria. The reviewers agreed in 93% of the studies on inclusion or exclusion, 7% of the articles were discussed. Reasons for exclusion were not meeting the criteria for: population (n = 150), intervention (n = 84), control (n = 1), type of study (n = 41), publication type (n = 7) or language (n = 1). Of the excluded studies, 24 investigated effectiveness of instructions and feedback with EF in children's functional gross motor learning in comparison with an IF and/or no instructions or feedback, without distinction in frequency, timing or form between groups [27–30,49–68]. Of the studies that distinguished in frequency, timing or form between groups, eight used an IF [69–76]. One study was excluded because its control group also used reduced instead of continuous frequency [77] (S2 File: overview of the excluded studies that nearly met inclusion criteria). Additionally, five studies were found through the references check, resulting in a total of 13 included studies (Fig 1).

### Methodological quality

Twelve RCTs were assessed with the RoB2, all of which having an overall RoB judgement of high [41,78–88] (Fig 2A). Percentages of agreements between reviewers varied (Domain 1: 75%; Domain 2: 25%; Domain 3: 41%; Domain 4: 25%; Domain 5: 67%). Although studies mentioned randomized groups, none described the generation method used and whether allocation was concealed [41,78–88]. Only one study provided a demographic characteristics table [79]. Most studies were at high risk for performance bias, none of the studies reported using intention-to-treat (ITT) analysis and how they handled missing data [41,78–88]. Most studies were also at high risk for detection bias, only one study reported no missing data [80]. In six studies, the F statistics showed that there were missing data, but information on the amount, at which time point and in which group was lacking [78,81,82,85–87]. In most studies, outcome assessors were not blinded or it remained unclear whether they were blinded [41,78,80–89]. None referred to a registered trial protocol, raising concerns about possible reporting bias [41,78–88]. The study of Hemayattalab & Rostami (2010) [89] was the only non-randomized CCT included. It had an overall judgement of serious RoB due to a serious RoB in measurement of outcomes, while the remaining domains were at low RoB [89] (Fig 2B). Reviewers scored similar for all domains except Domain 6.

### Study characteristics

Seven out of 13 studies included 348 inexperienced TDC [41,78,81–85], ages ranging from 6 [83,84] to 13 years [85]. Seven studies included 195 inexperienced children with motor

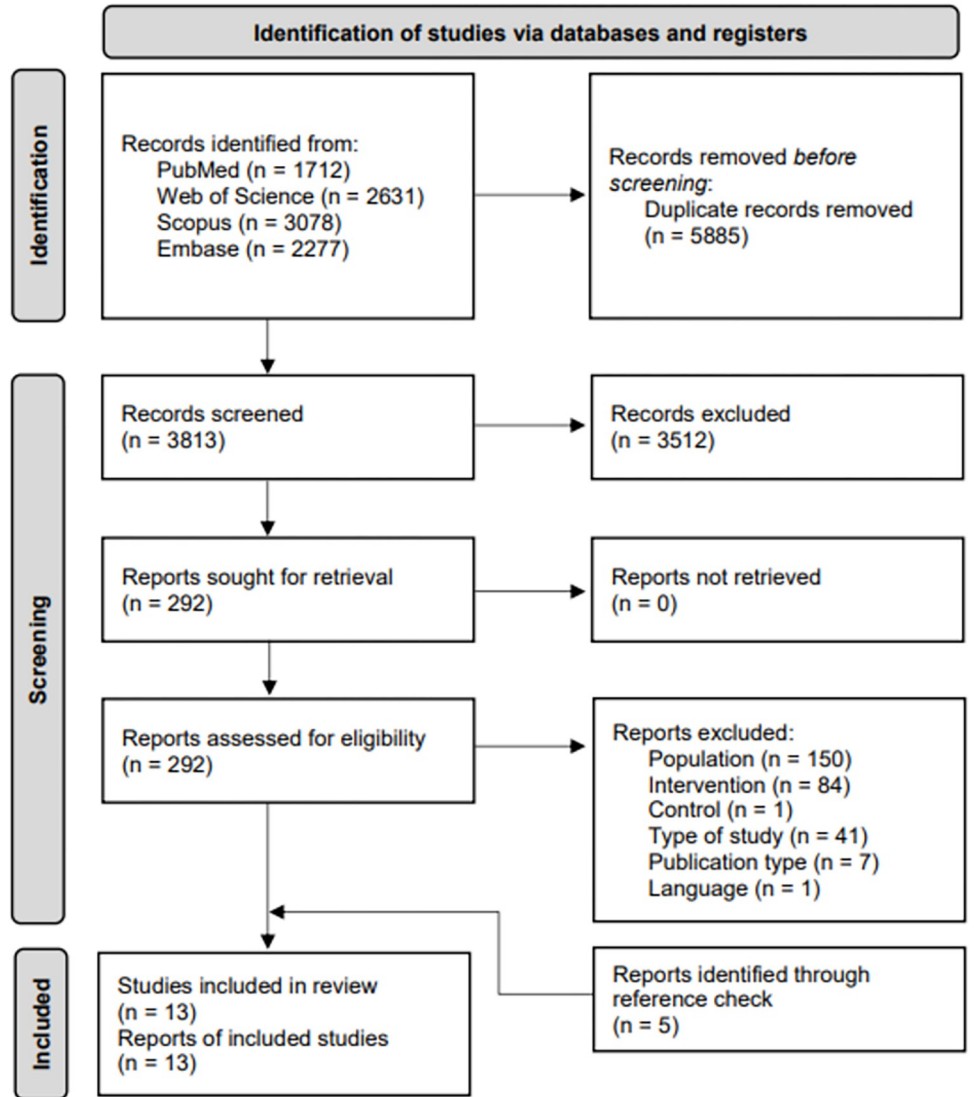

**Fig 1. Prisma flow diagram of the study selection.** N = number.

disabilities [79,80,83,86–89], ages ranging from 6 [79,83,87] to 18 years [87]. Mean ages and SDs were not reported in five studies [80,82–84,89]. The children with motor disabilities comprised children with MID [88], DCD [80], ASD [79,83] or CP [86,87,89]. Overall, the studies involved small sample sizes, the number of participants per group ranging from 6 [80] to 16 [88], with six studies having samples of 10 or less [78,80,83,86,87,89]. All studies used object control tasks [41,78–89]; 12 throwing [41,78–87,89] and one golf-putting [88]. In 10 studies, participants practised only once [41,78,79,81–86,88], the number of trials ranging from 30 [82] to 90 [79,85]. Participants in the remaining studies practised five times with a total of 100 trials [87], or eight times with a total of 240 trials [80,89]. All groups showed within group improvements during practice in 12 out of 13 studies [41,78–87,89] (Table 1).

The effectiveness of feedback with EF applied in **reduced frequency** compared to **continuous frequency** was investigated in eight studies [78,81–85,88,89], six of which included TDC [78,81–85]. The remaining studies included children with ASD [83] or CP [89]. The reduced

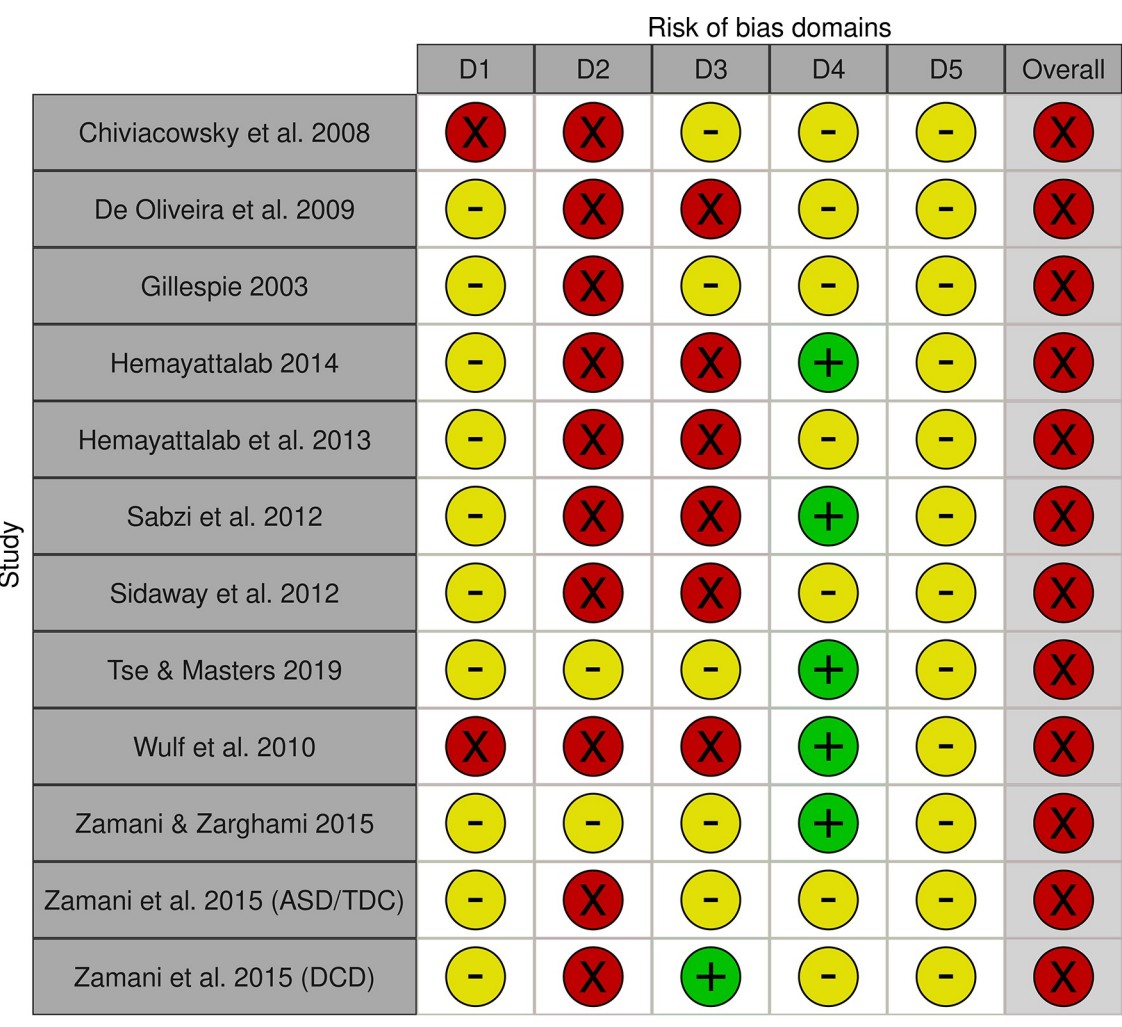

**Fig 2. Methodological quality of the included studies. a. Methodological quality assessed with RoB2.** D1 = selection bias; D2 = performance bias; D3 = detection bias; D4 = attrition bias; D5 = reporting bias; (+) = low risk; (-) = some concerns; (x) = high risk. **b. Methodological quality assessed with ROBINS-I.** D1 = bias due to confounding; D2 = selection bias; D3 = classification bias, D4 = bias due to deviation from intended interventions; D5 = bias due to missing data; D6 = bias in measurement of outcomes; D7 = reporting bias; (+) = low risk; (x) = serious risk.

frequency was applied in three fixed frequencies of 20% [88], 33% [81,82] and 50% [83–85,89], and one faded frequency decreasing from 100% to 0% with an average of 62% [78]. All studies assessed accuracy [78,81–85,88,89], with two also measuring variability [81,85], and one quality of movement [82]. Acquisition was assessed in all studies [78,81–85,88,89], while retention tests were used in seven [78,81–84,88,89], in which timing varied from 24 hours [78,82–84,88] to 1 week [81,88]. Only three studies measured transfer [81,82,85], in which timing varied from immediately after practice (0 hours) [81,82,85] to 1 week [81] (Table 1).

**Table 1. Study characteristics of the included studies.**

| Author(s), year | Population characteristics Type (n) Mean age in years (SD) Skill level | Study design | Intervention characteristics Groups (n) The provided instructions or feedback | Task | Practice | Measurements Outcome Covariables | Assessment time points | Results > significantly better than = no significant differences |
|---|---|---|---|---|---|---|---|---|
| **Frequency** | | | | | | | | |
| Gillespie, 2003 [88] | Children with MID, IQ range 55–70 (n = 32) 10.8 (0.68) Inexperienced | RCT | Groups • 20% frequency: KR after every fifth trial[a] • 100% frequency: KR after every trial[a] The provided feedbackthe instructor informed the child about the accuracy score and the child was allowed to see where the ball had stopped | Golf-putting task | 1 practice session of 10 blocks of 5 trials (total 50 trials) | Outcome Accuracy: score varied from 0–5 based on zones around the target (higher is better) | During acquisition Retention: • 24h • 1w | ANOVA analysis • During acquisition: No significant main effect for blocks Significant main effect for frequency 100% > 20* • Retention 24h: Significant main effect for frequency 20% > 100%* • Retention 1w: Significant main effect for frequency 20% > 100%* |
| Hemayattalab & Rostami, 2010 [89] | Children with CP, GMFCS 1 (n = 24) 7–15[b] Inexperienced | Non-rando-mized CCT | Groups • 50% frequency: KR on half of the trials (n = 8) • 100% frequency: KR after every trial (n = 8) • 0% frequency: no KR (n = 8) The provided feedback the instructor informed the child about the accuracy score | Dart throwing | 8 practice sessions of 6 blocks of 5 trials, period not reported (total 240 trials) | Outcome Accuracy: registered in points based on zones around the target (higher is better) | Pre-test During acquisition Post-test Retention: 72h | Significant paired sample t-tests for pre-test / post-test for all groups, all groups improved accuracy ANOVA analysis with post-hoc Tukey HSD tests Post-test: Significant main effect for frequency • Post-hoc testing: 100% > 50%**, 100% > 0%** and 50% > 0%* • Retention: Significant main effect for frequency Post-hoc testing: 50% > 100%**, 50% = 0% (p = 0.093) and 100% = 0% (p = 0.146) |
| de Oliveira et al., 2009 [85] | TDC (n = 120) 11.8 (1.2) Inexperienced | RCT | Groups • Simple task + 25% frequency: KR after a quarter of the trials (n = 15) • Simple task + 50% frequency: KR after half of the trials (n = 15) • Simple task + 75% frequency: KR after three quarters of the trials (n = 15) • Simple task + 100% frequency: KR after every trial (n = 15) • Complex task + 25% frequency: KR after a quarter of the trials (n = 15) • Complex task + 50% frequency: KR after half of the trials (n = 15) • Complex task + 75% frequency: KR after three quarters of the trials (n = 15) • Complex + 100% frequency: KR after every trial (n = 15) The provided feedback the child was allowed to see where the ball had stopped | Bocha game throwing • Simple task: throw with backward-forward pendulum movement of the arm • Complex task: throw with same pendulum movement followed by an overhead circular movement of the arm | 1 practice session of 9 blocks of 10 trials (total 90 trials) | Outcome[c] Accuracy: absolute error in cm (lower is better) Variability: variable error in cm (lower is better) | During acquisition Transfer: 0h | ANOVA analysis with post-hoc Tukey HSD tests Accuracy • During acquisition: Significant main effect for blocks, accuracy improved during practice Significant main effect for frequency Post-hoc testing: 25% > 50%, 75% and 100%* and 75% > 50%* • Transfer: Significant main effect for frequency Post-hoc testing: 25% > 50%* Variability • During acquisition: Significant main effect for blocks, variability decreased during practice Significant main effect for frequency Post-hoc testing: 25% > 50%, 75% and 100%* and 75% > 50% and 100%* • Transfer: Significant main effect for frequency Post-hoc testing: 25% > 50%, 75% and 100%* |

(Continued)

**Table 1.** (Continued)

| Author(s), year | Population characteristics Type (n) Mean age in years (SD) Skill level | Study design | Intervention characteristics Groups (n) The provided instructions or feedback | Task | Practice | Measurements Outcome Covariables | Assessment time points | Results > significantly better than = no significant differences |
|---|---|---|---|---|---|---|---|---|
| Sabzi et al., 2012 [78] | TDC (n = 40) 10.4 (1.0) Inexperienced | RCT | Groups • Reduced frequency: KR with faded frequency from 100% to 0% (n = 10) • 100% frequency: KR after every trial (n = 10) • Self-controlled: KR on request in 3 out of every 10 trials (n = 10) • Bandwidth: feedback when score is less than 50 out of 100 points (n = 10) The provided feedback the instructor informed the child about the results in terms of the direction and distance from the target centre | Throw with beanbag at target on the floor[d] | 1 practice session of 6 blocks of 10 trials (total 60 trials) | Outcome Accuracy: score varied from 0–100 based on zones around the target (higher is better) | During acquisition Retention: 24h | ANOVA analysis with post-hoc Tukey HSD tests • During acquisition: Significant main effect for blocks, accuracy increased during practice No significant main effect for groups • Retention: Significant main effect for groups Post-hoc testing: 100% > reduced feedback, SC and bandwidth* |
| Sidaway et al., 2012 [81] | TDC (n = 48) 10.7 (0.6) Inexperienced | RCT | Groups • Simple task + 33% frequency: KR after every third trial[a] • Simple task + 100% frequency: KR after every trial[a] • Complex task + 33% frequency: KR after every third trial[a] • Complex task + 100% frequency: KR after every trial[a] The provided feedbackthe instructor informed the child about the accuracy score | Throw with beanbag at target on the floor • Simple task: throw while standing • Complex task: throw while walking | 1 practice session of 6 blocks of 12 trials (total 72 trials) | Outcome[c] Accuracy: absolute error score varied from 0–3 based on zones around the target (lower is better) Variability: variable error based on the standard deviation about the mean score (lower is better) | During acquisition Post-test: in analysis referred to as retention 0h Retention: 1w Transfer: • 0h • 1w | ANOVA analysis[e] Accuracy • During acquisition: Significant main effect for blocks, accuracy increased during practice • Retention 0h and 1w: No significant main effect for frequency • Transfer 0h and 1w: No significant main effect for frequency Variability • During acquisition: Significant main effect for blocks, variability decreased during practice No significant main effect for frequency • Retention 0h and 1w: No significant main effect for frequency • Transfer 0h and 1w: No significant main effect for frequency |

*(Continued)*

**Table 1.** (Continued)

| Author(s), year | Population characteristics Type (n) Mean age in years (SD) Skill level | Study design | Groups (n) The provided instructions or feedback | Task | Practice | Outcome Covariables | Assessment time points | Results > significantly better than = no significant differences |
|---|---|---|---|---|---|---|---|---|
| | | | **Intervention characteristics** | | | **Measurements** | | |
| Wulf et al., 2010 [82] | TDC (n = 48) 10-12[b] Inexperienced | RCT | Groups • 33% frequency with EF: feedback after every third trial with focus on ball, target or shoes (n = 12) • 100% frequency with EF: feedback after every trial with focus on ball, target or shoes (n = 12) • 33% frequency with IF: feedback after every third trial with focus on the body (n = 12) • 100% frequency with IF: feedback after every trial with focus on the body (n = 12) The provided feedback • EF-feedback: example, *"The sneaker should point at the target; keep them apart"* • IF-feedback: example, *"The feet, hips, knees and shoulders should be aimed at the target, feet shoulder-width apart"* | Throw with soccer ball at target on the floor | 1 practice session of 6 blocks of 5 trials (total 30 trials) | Outcome[c] Accuracy: score varied from 0-5 based on zones around the target (higher is better) Quality of movement: score varied from 0-8 (8 criteria, per criteria 1 point if performed correctly and 0 point if not) (higher is better) | During acquisition Post-test: in results referred to as retention 0h Retention: 24h Transfer: • 0h • 24h | ANOVA analysis[e] Accuracy • During acquisition: Significant main effect for blocks, accuracy increased during practice No significant main effect for frequency • Retention: No significant main effect for frequency • Transfer: No significant main effect for frequency Quality of movement • During acquisition: Significant main effect for blocks, quality of movement increased during practice No significant main effect for frequency • Retention: No significant main effect for frequency Transfer: No significant main effect for frequency |
| Zamani & Zarghami, 2015 [84] | TDC (n = 45) 6-8[b] Inexperienced | RCT | Groups • 50% frequency: KR after every second trial[a] • 100% frequency: KR after every trial[a] • 0% frequency: no KR[a] The provided feedback the instructor informed the child about the results in terms of the direction and distance from the target centre | Throw with beanbag at target on the floor[d] | 1 practice session of 6 blocks of 10 trials (total 60 trials) | Outcome Accuracy: score varied from 0-100 based on zones around the target (higher is better) | Pre-test During acquisition Retention: 24h | ANOVA analysis with post-hoc Tukey HSD tests • During acquisition: Significant main effect for blocks, accuracy increased during practice Significant main effect for frequency Post-hoc testing: 100% > 50%**, 100% > 0%** and 50% > 0%** • Retention: Significant main effect for frequency Post-hoc testing: 50% > 100%*, 50% > 0%** and 100% > 0%** |

*(Continued)*

Table 1. (Continued)

| Author(s), year | Population characteristics Type (n) Mean age in years (SD) Skill level | Study design | Intervention characteristics Groups (n) The provided instructions or feedback | Task | Practice | Measurements Outcome Covariables | Assessment time points | Results > significantly better than = no significant differences |
|---|---|---|---|---|---|---|---|---|
| Zamani et al., 2015 [83] | TDC (n = 21) Children with ASD (n = 21) 6–8[b] Inexperienced | RCT | Groups • 50% frequency: KR after every second trial • (TDC: n = 7 / ASD: n = 7) • 100% frequency: KR after every trial • (TDC: n = 7 / ASD: n = 7) • 0% frequency: no KR (TDC: n = 7 / ASD: n = 7) The provided feedback the instructor informed the child about the results in terms of the direction and distance from the target centre | Throw with beanbag at target on the floor[d] | 1 practice session of 6 blocks of 10 trials (total 60 trials) | Outcome Accuracy: score varied from 0–100 based on zones around the target (higher is better) | Pre-test During acquisition Retention: 24h | ANOVA analysis with Tukey-Kramer tests TDC • During acquisition: Significant main effect for blocks, accuracy increased during practice Significant main effect for frequency Post-hoc testing: 100% > 50%**, 100% > 0%** and 50% > 0%** • Retention: Significant main effect for frequency Post-hoc testing: 50% > 100%**, 50% > 0%** and 100% > 0%** Children with ASD • During acquisition: Significant main effect for blocks, accuracy increased during practice Significant main effect for frequency Post-hoc testing: 100% > 50%**, 100% > 0%** and 50% > 0%** • Retention: Significant main effect for frequency Post-hoc testing: 100% > 50%**, 100% > 0%** and 50% > 0%** |

**Timing**

| Author(s), year | Population characteristics Type (n) Mean age in years (SD) Skill level | Study design | Intervention characteristics Groups (n) The provided instructions or feedback | Task | Practice | Measurements Outcome Covariables | Assessment time points | Results > significantly better than = no significant differences |
|---|---|---|---|---|---|---|---|---|
| Chiviacowsky et al., 2008 [41] | TDC (n = 26) 10.0 (0.5) Inexperienced | RCT | Groups • Self-controlled: KR on request (n = 13) • Yoked: KR whenever counterpart requested feedback (n = 13) The provided feedback the instructor informed the child about the results in terms of the direction and distance from the target centre | Throw with beanbag at target on the floor | 1 practice session of 6 blocks of 10 trials (total 60 trials) | Outcome Accuracy: score varied from 0–100 based on zones around the target (higher is better) | During acquisition Retention: 24h | ANOVA analysis • Acquisition: Significant main effect for blocks, accuracy increased during practice No significant main effect for timing • Retention: Significant main effect for timing SC > yoked* |
| Hemayattalab et al., 2013 [86] | Children with CP, GMFCS 1–3 (n = 24) 11.6 (1.5) Inexperienced | RCT | Groups • Self-controlled: KR on request (n = 10) • Yoked: KR whenever counterpart requested feedback (n = 10) The provided feedback the instructor informed the child about the results in terms of the direction and distance from the target centre | Throw with beanbag at target on the floor[d] | 1 practice session of 10 blocks of 8 trials (total 80 trials) | Outcome Accuracy: score varied from 0–100 based on zones around the target (higher is better) | Pre-test During acquisition Post-test Retention: 24h Transfer: 24h | A graph showed that both groups improved accuracy during practice. However, this was not tested with paired sample t-tests pre-test / post-test MANOVA analysis Post-test: • No significant main effect for timing (p = 0.473) Retention: • Significant main effect for timing (p = 0.003) SC > yoked Transfer: • Significant main effect for timing (p = 0.018) SC > yoked |

(Continued)

**Table 1.** (Continued)

| Author(s), year | Population characteristics Type (n) Mean age in years (SD) Skill level | Study design | Intervention characteristics Groups (n) The provided instructions or feedback | Task | Practice | Measurements Outcome Covariables | Assessment time points | Results > significantly better than = no significant differences |
|---|---|---|---|---|---|---|---|---|
| Hemayattalab et al., 2014 [87] | Children with CP, GMFCS 1 (n = 22) 12.26 (3.11) Inexperienced | RCT | Groups • Self-controlled: KR on request in maximum half of the trials (n = 8) • Instructor-controlled: KR when instructor wanted in half of the trials (n = 7) • Control: no KR (n = 7) The provided feedback the instructor informed the child about the accuracy score | Dart throwing | 5 practice sessions of 4 blocks of 5 trials, period not reported (total 100 trials) | Outcome Accuracy: score varied from 0–100 based on zones around the target (higher is better) | During acquisition (mean scores practice session 1 and 5 used as pre- and post-test) Retention: 24h Transfer: 24h | ANOVA analysis with post-hoc Tukey HSD tests • During acquisition: Significant main effect for days, accuracy increased during practice No significant main effect for timing • Retention: Significant main effect for timing Post-hoc testing; SC = IC*, SC > control (p = 0.014) and IC = control* • Transfer: Significant main effect for timing Post-hoc testing; SC = IC*, SC > control* and IC = control (p > 0.05) |
| Sabzi et al., 2012 [78] | TDC (n = 40) 10.4 (1.0) Inexperienced | RCT | Groups • Reduced frequency: KR with faded frequency from 100% to 0% (n = 10) • 100% frequency: KR after every trial (n = 10) • Self-controlled: KR on request in 3 out of every 10 trials (n = 10) • Bandwidth: feedback when score is less than 50 out of 100 points (n = 10) The provided feedback the instructor informed the child about the results in terms of the direction and distance from the target centre | Throw with beanbag at target on the floor[d] | 1 practice session of 6 blocks of 10 trials (total 60 trials) | Outcome Accuracy: score varied from 0–100 based on zones around the target (higher is better) | During acquisition Retention: 24h | Anova analysis with post-hoc Tukey HSD tests • During acquisition: Significant main effect for blocks, accuracy increased during practice No significant main effect for groups • Retention: Significant main effect for groups Post-hoc testing: 100% > reduced feedback, SC and bandwidth* |
| Zamani et al., 2015 [80] | Children with DCD and MID, IQ range 50–70 (n = 24) 9-11[b] Inexperienced | RCT | Groups • Self-controlled + 75% frequency: KR on request in 75% of the trials (n = 6) • Self-controlled + 50% frequency: KR on request in half of the trials (n = 6) • Instructor-controlled + 75% frequency: KR when instructor wanted in 75% of the trials (n = 6) • Instructor-controlled + 50% frequency: KR when instructor wanted in half of the trials (n = 6) The provided feedback the instructor informed the child about the results in terms of the direction and distance from the target centre | Throw with tennis ball at target[f] (unclear whether target is placed on floor or wall) | 8 practice sessions of 6 blocks of 10 trials, period not reported (total 240 trials) | Outcome Accuracy: score varied from 0–100 based on zones around the target (higher is better) | Pre-test During acquisition Post-test Retention: timing unclear | Significant paired sample t-tests for pre-test / post-test for all groups, all groups improved accuracy Anova analysis with post-hoc Tukey HSD tests • Post-test: No significant main effect for timing • Retention: Significant main effect for timing SC > IC** |

Form

(Continued)

**Table 1.** (Continued)

| Author(s), year | Population characteristics<br>Type (n)<br>Mean age in years (SD)<br>Skill level | Study design | Intervention characteristics<br>Groups (n)<br>The provided instructions or feedback | Task | Practice | Measurements<br>Outcome Covariables | Assessment time points | Results<br>> significantly better than<br>= no significant differences |
|---|---|---|---|---|---|---|---|---|
| Tse & Masters, 2019 [79] | Children with ASD and MID, IQ range 50–70<br>(n = 48)<br>10,10 (2.0)<br>Inexperienced | RCT | Groups<br>• Instruction with visual analogy (n = 12)<br>• Instruction with verbal analogy (n = 12)<br>• Explicit instruction: with IF on arm or hand (n = 12)<br>• Control: no specific instruction (n = 12)<br><u>The provided instructions</u><br>• Visual analogy: an illustration of a child putting a cookie in a cookie jar on a high shelf<br>• Verbal analogy: *"Shoot the ball as if you are trying to put cookies into a cookie jar on a high shelf"*<br>• Explicit instruction: *"Move the ball upward and release the ball when your strong arm becomes vertical"*; *"When releasing the ball, your strong hand is facing downwards"* | Basketball free throw | 1 practice session of 6 blocks of 15 trials<br>(total 90 trials) | <u>Outcome</u><br>Accuracy: score varied from 0–5 based on zones around the target (higher is better)<br><u>Covariables</u><br>Age<br>IQ<br>WMI<br>SRS-2 | During acquisition<br>Retention: 24h<br>Transfer: 24h | Pearson correlation<br>• Covariables were not correlated to accuracy score<br>ANOVA analysis with post-hoc tests with Bonferroni correction<br>• <u>During acquisition:</u><br>Significant main effect for blocks<br>Post-hoc testing: the 3 instruction groups improved performance during practice, while the control group did not<br>Significant main effect for group<br>Post-hoc testing: visual analogy > control**, verbal analogy > control** and explicit > control**<br>• <u>Retention:</u><br>Significant main effect for group<br>Post-hoc testing: visual analogy > verbal analogy** 95% CI = [1.57,9.77], visual analogy > explicit** 95% CI = [1.07,9.27], visual analogy > control** 95% CI = [5.65,13.85], verbal analogy > control* 95% CI = [-0.02,8.18], explicit > control* 95% CI = [0.48,8.68] and verbal analogy = explicit (p = 0.74) 95% CI [-4.60,3.60]<br>• <u>Transfer:</u><br>Significant main effect for group<br>Post-hoc testing: visual analogy > verbal analogy** 95% CI = [2.50,9.00], visual analogy > explicit** 95% CI = [1.41,7.92], visual analogy > control** 95% CI = [5.91,12.42], verbal analogy > control* 95% CI = [0.16,6.67], explicit > control* 95% CI = [1.25,7.75] and verbal analogy = explicit (p = 0.39) 95% CI [-4.34,2.17] |

N = number; SD = standard deviation; h = hour(s); w = week(s); RCT = randomized controlled trial; CCT = controlled clinical trial; KR = knowledge of results; EF = external focus; IF = internal focus; MID = mild intellectual disabilities; IQ = intelligent quotient; CP = cerebral palsy; GMFCS = gross motor functioning classification system; TDC = typically developing children; ASD = autistic spectrum disorder; DCD = developmental coordination disorder; SC = self-controlled; IC = instructor-controlled; CI = confidence interval; IQ = intelligence quotient; WMI = Working Memory Index; SRS-2 = social responsiveness scale, 2nd ed.

a number per group not reported

b only age range was reported

c primary and secondary outcome not specified

d according to protocol Chiviacowsky 2008 [41]

e not reported which post-hoc test used

f same target as Chiviacowsky 2008 [41]

* p < 0.05; ** p ≤ 0.001.

Effectiveness of **self-controlled feedback** compared to **instructor-controlled feedback** to improve accuracy in object control tasks was investigated in five studies [41,78,80,86,87]. TDC were included in two studies [41,78], while the others included children with DCD [80] or CP [86,87]. In four studies, the frequency of the self- and instructor-controlled feedback was the same [41,80,86,87], while in one frequencies were different, 30% in the self-controlled group and 100% in the instructor-controlled group [78]. All studies measured acquisition and retention [41,78,80,86,87]. In most studies, retention was measured after 24 hours [41,78,86,87], though in one the timing was unclear (80). One-day transfer tests were used in two studies [86,87] (Table 1).

One study with children with ASD and MID investigated the effectiveness of **visual analogy** compared to **verbal analogy** for improving accuracy in basketball shooting on acquisition, retention (24 hours), and transfer (0 and 24 hours) [79] (Table 1).

## Best-evidence synthesis

Regarding frequency of feedback (hypothesis 1), three out of seven studies investigated the effectiveness of reduced fixed frequency in similar tasks [81,83,84]. However, one reported no summary statistics [81] and the other two had the same first author [83,84]. The remaining studies used non-comparable tasks [82,85,88,89]. Only one study examined the effectiveness of reduced faded frequency. As regards timing of feedback (hypothesis 2), four out of five studies included similar tasks [41,78,80,86], but summary statistics were lacking in two of these [41,78]; the remainder included different populations [80,86], and only one investigated a visual form of instruction (hypothesis 3). Therefore, all studies were included in the best-evidence synthesis [41,78–89] (Table 2). Although each study described whether there were significant group differences, none mentioned thresholds for minimal clinically important differences [41,78–89].

The following paragraphs describe the results from the best-evidence synthesis for the parameters frequency, timing and form. For frequency, results were reported for the outcomes accuracy, variability and quality of movement. Studies of timing and form only assessed accuracy. For each parameter, results are ordered according to the following time points: 1. Acquisition measured during practice; 2. Acquisition measured with a post-test; 3. Retention; and 4. Transfer.

**Frequency.** The evidence whether reduced fixed frequency of feedback was more effective than continuous frequency (hypothesis 1) in improving accuracy of object control tasks on acquisition was conflicting [81,82,84,85,88,89]. For *acquisition measured during practice*, continuous frequency appeared more effective in TDC [83,84] and in children with ASD [83] or MID [88]; however, two other studies with TDC found no significant group differences [81,85]. For *acquisition measured with a post-test*, the results of the studies varied with the population. No significant group differences were found in TDC [81,82], while continuous frequency appeared more effective in children with CP [89]. For *retention*, conflicting evidence was also found [81,82,84,85,88,89]: for TDC, two studies found no significant group differences [81,82], while two other studies indicated that reduced frequency was more effective [83,84]; for children with motor disabilities, results showed that children with CP [89] and MID [88] performed best with reduced frequency while children with ASD did best with continuous frequency [83]. For *transfer*, no evidence supported either frequency in TDC [81,82,85] (Table 2). Only one study compared reduced faded frequency to continuous frequency to improve accuracy in beanbag throwing in TDC [78]. For *acquisition measured during practice*, they found no significant group differences [78]. For *retention*, limited evidence was found favouring continuous frequency [78] (Table 2).

**Table 2. Best-evidence synthesis of instructions or feedback applied with a specific frequency, timing or form.**

| Parameter studied | Comparison | Author | Task | Population | Evidence synthesis per study | | | | | | Evidence synthesis summary | | | |
|---|---|---|---|---|---|---|---|---|---|---|---|---|---|---|
| | | | | | Acquisition | | Retention | | Transfer | | Acquisition | | Retention | Transfer |
| | | | | | During | Post | Timing | Effect | Timing | Effect | During | Post | | |
| | | | | | *Accuracy* | | | | | | | | | |
| Frequency | Reduced fixed vs continuous | Sidaway et al. 2012 [81] | Throw with beanbag | TDC | NS | NS | 1w | NS | 0h 1w | NS NS | X | X | X | - |
| | | Zamani & Zarghami 2015 [84] | Throw with beanbag | TDC | C | NA | 24h | R | NA | NA | | | | |
| | | Wulf et al. 2010 [82] | Throw with soccer ball | TDC | NA | NS | 24h | NS | 0h 24h | NS NS | | | | |
| | | De Oliveira et al. 2009 [85] | Throw with bocha ball | TDC | NS | NA | NA | NA | 0h | NS | | | | |
| | | Zamani et al. 2015 [83] | Throw with beanbag | TDC | C | NA | 24h | R | NA | NA | | | | |
| | | | Throw with beanbag | Children with ASD | C | NA | 24h | C | NA | NA | | | | |
| | | Hemayattalab & Rostami 2010 [89] | Dart throwing | Children with CP | NA | C | 72h | R | NA | NA | | | | |
| | | Gillespie 2003 [88][a] | Golf putting | Children with MID | C | NA | 24h 1w | R R | NA | NA | | | | |
| | Reduced faded vs continuous | Sabzi et al. 2012 [78] | Throw with beanbag | TDC | NS | NA | 24h | C | NA | NA | - | NA | * C | NA |
| Timing | Self-controlled vs instructor-controlled (equal frequency in both groups) | Chiviacowsky et al. 2008 [41] | Throw with beanbag | TDC | NS | NA | 24h | SC | NA | NA | X | - | ** SC | X |
| | | Hemayattalab et al. 2013 | Throw with beanbag | Children with CP | NS | NA | 24h | SC | 24h | SC | | | | |
| | | Hemayattalab et al. 2014 [87] | Dart throwing | Children with CP | SC | NA | 24h | NS | 24h | NS | | | | |
| | | Zamani et al. 2015 [83] | Throw with tennis ball | Children with DCD and MID | NA | NS | NR | SC | NA | NA | | | | |
| | Self-controlled 30% vs instructor-controlled 100% feedback | Sabzi et al. 2012 [78] | Throw with beanbag | TDC | NS | NA | 24h | C | NA | NA | - | NA | * IC-100% | NA |
| Form | Visual analogy vs verbal analogy | Tse & Masters 2019 [79] | Basketball free throw | Children with ASD and MID | NS | NA | 24h | VisA | 24h | VisA | - | NA | * VisA | * VisA |
| | | | | | *Variability* | | | | | | | | | |
| Frequency | Reduced fixed vs continuous | Sidaway et al. 2012 [81] | Throw with beanbag | TDC | NS | NS | 1w | NS | 1w | NS | - | - | - | - |
| | | De Oliveira et al. 2009 [85] | Throw with bocha ball | TDC | NS | NA | NA | NA | 0h | NS | | | | |

(*Continued*)

**Table 2.** (Continued)

| Parameter studied | Comparison | Author | Task | Population | Evidence synthesis per study | | | | | | Evidence synthesis summary | | | |
|---|---|---|---|---|---|---|---|---|---|---|---|---|---|---|
| | | | | | Acquisition | | Retention | | Transfer | | Acquisition | | Retention | Transfer |
| | | | | | During | Post | Timing | Effect | Timing | Effect | During | Post | | |
| **Quality of movement** | | | | | | | | | | | | | | |
| Frequency | Reduced fixed vs continuous | Wulf et al. 2010 [82] | Throw with soccer ball | TDC | NA | NS | 24h | NS | 0h<br>24h | NS<br>NS | NA | - | - | - |

[a] = the groups did not improved performance during practice; h = hour(s); w = week(s); TDC = typically developing children; ASD = autistic spectrum disorder; CP = cerebral palsy; MID = mild intellectual disabilities; DCD = developmental coordination disorder; NA = not applicable; NR = not reported; C = significant, favouring continuous frequency; R = significant, favouring reduced frequency; SC = significant, favouring self-controlled feedback; IC-100% = significant, favouring instructor-controlled feedback after every trial; VisA = significant, favouring the visual analogy. Consistency was defined as 75% of the studies assessing the same comparison showing results in the same direction. Strength of the evidence according to the guidelines of van Tulder et al.:

*** = Strong–consistent findings among multiple high quality RCTs.

** = Moderate–consistent findings among multiple low quality RCTs and/or CCTs and/or one high quality RCT.

* = Limited–one low quality RCT and/or CCT.

X = conflicting–inconsistent findings among multiple RCTs and/or CCTs.

- = no evidence from trials–no RCTs or CCTs.

There was no evidence that reduced fixed or continuous frequency was more effective in reducing **variability** or improving **quality of movement** in throwing in TDC for *acquisition, retention and transfer* [81,82,85] (Table 2).

**Timing.** For accuracy in object control tasks, conflicting evidence was found on effectiveness of self-controlled versus instructor-controlled feedback (hypothesis 2) with equal frequency for *acquisition measured during practice* [41,86,87]. Of the studies including children with CP [86,87], one showed that self-controlled timing was more effective [87], while another found no significant group differences [86]; no significant group differences were found in TDC [41]. Also, no significant group differences were found in children with DCD for *acquisition measured with a post-test* [80]. For *retention*, the self-controlled group performed best in three studies [41,80,86], including TDC [41], children with CP [86] and DCD [80]. A fourth study showed no significant group differences in children with CP [87], which resulted in only moderate evidence favouring self-controlled timing [41,80,86,87]. For *transfer*, the evidence was conflicting in children with CP: while one study showed that self-controlled timing was more effective, another found no significant group differences [86,87] (Table 2).

One study used different frequencies in the self- and instructor-controlled groups to improve **accuracy** in beanbag throwing in TDC [78]. For *acquisition measured during practice*, no evidence supported either timing. However, there was limited evidence that 100% instructor-controlled feedback was more effective than 30% self-controlled feedback for *retention* [78] (Table 2).

**Form.** One study investigated the effectiveness of visual analogy compared to verbal analogy (hypothesis 3) used to improve accuracy in basketball throwing in children with ASD and MID (79). For *acquisition measured with a post-test*, no evidence supported either form [79]. However, for *retention* limited evidence was found favouring a visual form of instruction [79] (Table 2).

## Discussion

The aim of this systematic review was to investigate the effectiveness of instructions and feedback with EF applied with reduced frequency, with self-controlled timing or in visual form in

the learning by (a)typically developing children of functional gross motor tasks. Although, the constrained action hypothesis suggested that an EF would be more effective, previous research investigating effectiveness of instructions or feedback with EF found conflicting results for children [23,36] and adults [43,90]. It was hypothesized that the frequency, timing and/or form of instructions and feedback [20] influenced their effectiveness. The following paragraphs will discuss results by each hypothesis.

First, it was hypothesized that reduced frequency would be more effective than continuous frequency. However, the results of the best-evidence synthesis did not support this. For acquisition, conflicting evidence was found for accuracy, but studies found either no significant group differences [78,81,82,85] or significant differences favouring continuous frequency [83,84,88,89]. A possible reason why continuous frequency appeared more effective could be the short practice duration, as most studies included only one practice session [78,81–85,88] (Table 1). At the beginning of the learning process, feedback dependency is likely to be higher because more information (e.g. by means of more instructions and feedback) is needed to acquire new skills [12,34,91,92]. With inexperienced children, it is likely that some children remained in the early learning stage due to insufficient repetitions and, therefore, performed better with continuous frequency. In practical settings, children have longer training periods. Therefore, future studies adopting longer practice durations would be of more practical interest which will improve ecological validity as well. For retention, conflicting evidence was found for accuracy as well, however, four out of seven experiments found beneficial effects for reduced frequency [83,84,88,89] as expected [34]. From the remaining three studies, two found non-significant results [81,82]. For transfer, no evidence was found for accuracy [81,82,85]. However, these studies, also measuring variability and quality of movement, found non-significant results for acquisition and retention as well [81,82,85]. Only one study compared a faded reduced frequency to a continuous frequency in TDC using a one-day training protocol, resulting in limited evidence for continuous frequency for retention [78]. The interpretation of these results might be influenced due to methodological limitations, which will be elaborated later. This limited or conflicting evidence is in line with previous research. Systematic reviews investigating effectiveness of frequency of feedback to improve motor skills in TDC and children with CP found limited or contradicting evidence for children with CP [32,33]. They suggested that child characteristics and task complexity might moderate effectiveness, but foremost they recommended that more studies of methodologically sound quality including the investigation of relevant child characteristics are needed to draw conclusions [32,33]. For TDC, they concluded that reduced frequency might be more effective [33]. However, two studies investigating the effectiveness of reduced frequency in TDC and CP did not include a control group with a continuous frequency. Furthermore, the study that compared a continuous with a faded frequency found no differences between groups for TDC [33]. In summary, several individual studies in the best-evidence synthesis showed beneficial effects for reduced frequency for retention, and for continuous frequency for acquisition. However, overall results in this, and previous studies, were conflicting. Therefore, it was not possible to draw conclusions about the preferred frequency.

Secondly, it was hypothesized that self-controlled timing would be more effective than instructor-controlled timing. The results of the best-evidence synthesis confirmed this, with moderate evidence for retention when frequency of feedback was similar in both groups [41,80,86,87] (Table 2). On the contrary, when frequencies were dissimilar, the instructor-controlled group appeared more effective for retention [78]. This inconsistency may be due to the frequency of feedback, as the self-controlled group received less feedback than the instructor-controlled group during the one-day training protocol [78] (Table 1). For all other time points, either no or conflicting evidence was found. However, if results were conflicting, studies found

either non-significant results or evidence favouring self-controlled timing as was expected by the Self-Determination Theory [41,80,86,87]. The non-significant results might be due to the low methodological quality of the included studies, which will be elaborated later. In this study, the yoked and instructor-controlled groups were combined as control. However, it can be argued that effectiveness can differ depending on the type of control group. Moreover, instructor-controlled feedback may be more supportive to the child than the yoked controlled feedback because of its timing; it is to be expected that the instructor estimates when the feedback would be most informative to the child, while in the yoked condition the moment of feedback is not related to the child's performances. It would be interesting to explore this assumption in future research. A systematic review investigating the effectiveness of autonomy support in children's functional skill motor learning yielded similar results [36]. It found that self-controlled feedback was more effective in several studies, but it was argued that child characteristics, like trait anxiety, cognitive skills and age, may have influenced effectiveness [36]. In the best-evidence synthesis, three out of four studies with equal frequency of feedback in both groups included children with either CP [86,87] or DCD [80]. These children are characterized by cognitive deficits, which might influence their abilities for autonomous functioning [6,37,93]. These characteristics, in addition to the methodological limitations, might explain why results are not as consistent as expected [37]. Although more evidence is needed to draw conclusions for all time points, the results from the best-evidence synthesis, supported by previous research, suggests that instructors should consider using self-controlled timing in children's motor learning.

Finally, it was hypothesized that children learnt functional gross motor skills best with a visual form of instructions and feedback compared to a verbal form. However, only one study investigated this specific comparison [79]. Post-hoc comparisons showed that children with ASD threw more accurately after a visual analogy [79]. Similar results were found in studies with healthy young adults and young adults with Down syndrome, where skill performance improved more after video [94,95] or instructor demonstration [96] than with verbal instructions with EF. Although evidence is limited, instructors might consider using pictures, videos or real live demonstrations as instructions or feedback to teach children motor skills.

This was the first study to systematically investigate the modifying role of frequency, timing and form in instructions and feedback with EF on children's motor learning. A strength of this study was that it followed a registered protocol, comprising a selection process and RoB assessment performed by two reviewers independently, with an epidemiologist (CB) to be consulted in cases of disagreement. Furthermore, RoB was assessed by means of reference standards (the Cochrane RoB tools) and findings were analysed according to a prespecified plan. There was no need to contact authors of included studies for further details. There is a small possibility that we interpreted reported information slightly different than meant by the authors. This study included functional tasks which improved the ecological validity of this study.

Providing recommendations for instructors about the frequency, timing and form of instructions and feedback with EF appeared challenging for three particular reasons. Firstly, drawing evidence-based conclusions was difficult because of the poor methodological quality of the studies [41,78–88] (Fig 2). In particular, blinding of outcome assessors, analysing according to ITT, and handling missing data properly require attention in future studies [97,98]. Furthermore, authors should report methods and results in more detail, essential for adequately determining the RoB [97,98]. It is possible that methodological quality appeared lower due to insufficient reporting of details. Additionally, the generally small sample sizes and the lack of reported thresholds of clinically meaningful differences also hindered interpretation. Inadequate sample sizes increase the risk of finding non-significant results or contrary conclusions with similar studies [99,100]. This might have influenced the number of non-

significant results found in individual studies and, more specifically, the lack of evidence or the conflicting evidence in the best-evidence synthesis (Table 2) [99,101]. In particular, the results of the post-hoc comparisons should be interpreted cautiously [99]. Although some studies found significant differences, it remains unclear whether these differences are large enough to be relevant in practical settings [102,103]. More methodologically sound studies based on proper sample size calculations are needed to draw conclusions regarding the preferred frequency, timing and form of instructions and feedback.

Secondly, it is suggested that child and task characteristic may moderate effectiveness [23,36]. However, more research is necessary to gain insights into which characteristics are relevant, and their moderating role. Accordingly, it was not possible to perform sub analyses in the best-evidence synthesis. For instructors, it is not only important to know how to shape their instructions and feedback, but also how to adapt their instructions and feedback to child and task [17,104]. Therefore, performing sub analyses on all potentially relevant variables such as typical/atypical development, age, cognitive or motivational factors, would be recommended for future research when more methodologically sound studies are available, including relevant data to make sub groups properly.

Thirdly, generalizability of the results was hampered because all included studies used object control tasks with inexperienced children, and measured accuracy. This overrepresentation of tasks, skill level and outcome is in line with previous research [23,36]. In therapy, PE classes and sports, children learn various tasks with different levels of complexity [105] and, depending the child's needs, instructors teach new tasks to novice children or optimize existing skills in experienced or trained children [8,106,107]. The *challenge point framework* conceptualizes the amount and specificity of information needed to learn skills, based on the level of task complexity, the skill level of the individual, and the interaction of level of complexity with skill level [91]. This framework, and other studies, suggest that instructors should adapt frequency, timing and form of instructions and feedback to the individual and the task [17,23,36,91,104]. Child characteristics as skill level, cognitive functioning, motivation, and the presence of a diagnose are considered relevant [17,23,36,104]. However, more research is necessary to gain a better understanding of their moderating role. Therefore, future research should attempt to include a wider variety of tasks and/or child characteristics in their studies. This will improve ecological validity, and generalizability of the studies as well. In order to guarantee comparability of studies, a framework that classifies tasks based on their characteristics could be helpful. Future research should give attention to developing such a framework. Potentially relevant characteristics are the number of degrees of freedom, cognitive demands, sequence of movement structure, spatial and temporal demands, and the context of tasks [2,47,92]. As for outcome, few studies assessed variability [81,85] or quality of movement [82], as well as accuracy. In practical settings, instructors often focus on improving functionality instead of normality [8,106,107]. From that point of view, accuracy is a relevant outcome, because it focuses on the result of the performance instead of on the optimal movement pattern. However, instructors can target various improvements, depending on the child's need. Therefore, for better ecological validity, more result-related outcomes (e.g. variability, number of successful attempts and distance) and movement pattern-related outcomes (e.g. quality of movement and kinematic variables) should be considered in future studies. Irrespective of the chosen outcome, researchers should use valid, reliable and responsive outcome measures.

## Conclusion

Based on the results of this systematic review, instructors should consider using self-controlled feedback with EF to enhance children's motor learning (moderate evidence). Regarding a

specific frequency or form, no conclusions can be drawn yet. However, based on limited evidence, instructors could consider using visual instructions. Because specific child and task characteristics can also moderate the effectiveness of instructions and feedback [23,36,91], instructors should explore the optimal frequency, timing and form for each child until more research provides us with a better understanding of their moderating role. Future research should put effort into developing a framework that classifies tasks based on their characteristics. Furthermore, it should aim to advance insights into the modifying role of frequency, timing and form in instructions and feedback with EF with methodologically sound studies focusing on: 1. a variety of tasks; 2. populations with different skill levels, age ranges, and diagnoses; 3. various outcome measures; and 4. with longer practice duration.

## Supporting information

**S1 Checklist.**
(DOCX)

**S1 File. Search queries for the individual databases.**
(DOCX)

**S2 File. Excluded studies that nearly met inclusion criteria.**
(DOCX)

## Author Contributions

**Conceptualization:** Ingrid P. A. van der Veer, Evi Verbecque, Eugene A. A. Rameckers, Caroline H. G. Bastiaenen, Katrijn Klingels.

**Data curation:** Ingrid P. A. van der Veer, Evi Verbecque.

**Formal analysis:** Ingrid P. A. van der Veer, Evi Verbecque.

**Investigation:** Ingrid P. A. van der Veer, Evi Verbecque, Eugene A. A. Rameckers, Katrijn Klingels.

**Methodology:** Ingrid P. A. van der Veer, Evi Verbecque, Eugene A. A. Rameckers, Caroline H. G. Bastiaenen, Katrijn Klingels.

**Supervision:** Eugene A. A. Rameckers, Caroline H. G. Bastiaenen, Katrijn Klingels.

**Writing – original draft:** Ingrid P. A. van der Veer.

**Writing – review & editing:** Ingrid P. A. van der Veer, Evi Verbecque, Eugene A. A. Rameckers, Caroline H. G. Bastiaenen, Katrijn Klingels.

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
