## [Decision Letter · Decision Letter 0]

6 Jun 2022

PONE-D-22-04840How can instructions and feedback with external focus be shaped to enhance motor learning in children? A systematic reviewPLOS ONE

Dear Dr. van der Veer,

Thank you for submitting your manuscript to PLOS ONE. After careful consideration, we feel that it has merit but does not fully meet PLOS ONE’s publication criteria as it currently stands. Therefore, we invite you to submit a revised version of the manuscript that addresses the points raised during the review process.

ACADEMIC EDITOR:Dear Authors, two experts in the field reviewed your manuscript and reported some major points you should consider in the next step of the revision process.

We look forward to receiving your revised manuscript.

Kind regards,

Emiliano Cè

Academic Editor

PLOS ONE

**Journal requirements:**

Reviewers' comments:

Reviewer's Responses to Questions

**Comments to the Author**

1. Is the manuscript technically sound, and do the data support the conclusions?

Reviewer #1: Partly

Reviewer #2: Partly

2. Has the statistical analysis been performed appropriately and rigorously? 

Reviewer #1: N/A

Reviewer #2: N/A

3. Have the authors made all data underlying the findings in their manuscript fully available?

Reviewer #1: Yes

Reviewer #2: Yes

4. Is the manuscript presented in an intelligible fashion and written in standard English?

Reviewer #1: Yes

Reviewer #2: Yes

5. Review Comments to the Author

Reviewer #1: This paper presents the results of a systematic review on the effects of instructions and feedback with an external focus of attention on motor learning in children. It focusses on the manipulation of frequency, timing and form and its effects on the performance of functional skills. Based on the best-evidence synthesis, the authors cautiously conclude that there is some evidence for using self-controlled feedback, visual instructions and a continuous frequency of feedback. Improving motor skill learning in both typically and a-typically developing children is both a scientifically and practically relevant topic. Therefore, the review can potentially contribute to the field. However, I do have some questions and suggestions, mostly related to the theoretical frameworks and interpretations of the findings and with that also question the conclusions that are drawn. These will be listed in the attachment

Reviewer #2: This systematic review is focused on the role played by frequency, timing and form on the performance of functional skills. The authors reported some possible evidence for using self-controlled feedback, visual instructions and a continuous frequency of feedback. The review deals with an interesting topic and has some potentialities. I have only some concerning about the Authors should consider.

The introduction is focused and well-written, the scope is clear, and the bibliography updated.

The methods are well-structed and the procedure used are accurate. I have only a concern about the merging results from children with a typical and an atypical development. This should be considered and discussed.

The results are well reported

The discussion is well supported by the data provided by the meta-analysis. As above stated, I would reconsider the part coming from the combination of study including typically and atypically developing children.

6. PLOS authors have the option to publish the peer review history of their article (what does this mean?). If published, this will include your full peer review and any attached files.

Reviewer #1: **Yes: **Dr. Femke van Abswoude

Reviewer #2: No

---

## [Author Response · Author response to Decision Letter 0]

21 Jul 2022

Author’s response to decision letter for PONE-D-22-04840 (Also uploaded as separate document)

How can instructions and feedback with external focus be shaped to enhance motor learning in children? A systematic review  

Diepenbeek, 19-07-2022

Dear editor in chief, dear Prof. E. Cè, 

Please find uploaded our revision of the manuscript entitled: “How can instructions and feedback with external focus be shaped to enhance motor learning in children? A systematic review.” 

We revised our manuscript as requested in your email of June 6, 2022.

We would like to thank the editor and the reviewers for their constructive and detailed feedback, and for giving us the opportunity to revise our manuscript. We have implemented most of the reviewers’ comments and suggestions. However, after careful consideration, we have decided not to perform analyses with sub groups of typical and atypical developing children. Below we provide a point-to-point reply to each of the comments (in italic blue colored). In the manuscript, we highlighted the additions in the manuscript by coloring the text blue and used track-changes for the deleted text. (The pages and lines referred to in the point-to-point reply apply to the manuscript with the track-changes. The lines of the manuscript with and without track-changes appeared not similar, but we have not been able to solve this problem. Our excuses for the inconveniences.)

We hope that with this revision and reply all concerns are satisfactorily addressed and the manuscript can be accepted for publication. Of course, we will be happy to answer any additional questions from the editorial office or reviewers.

Yours sincerely,

Ingrid van der Veer (first author, on behalf of all co-authors)

 

Reviewer comments:

Reviewer #1

This paper presents the results of a systematic review on the effects of instructions and feedback with an external focus of attention on motor learning in children. It focusses on the manipulation of frequency, timing and form and its effects on the performance of functional skills. Based on the best-evidence synthesis, the authors cautiously conclude that there is some evidence for using self-controlled feedback, visual instructions and a continuous frequency of feedback. Improving motor skill learning in both typically and a-typically developing children is both a scientifically and practically relevant topic. Therefore, the review can potentially contribute to the field. However, I do have some questions and suggestions, mostly related to the theoretical frameworks and interpretations of the findings and with that also question the conclusions that are drawn. These will be listed in the attachment (copied below).

Major issues/comments

Introduction

1. The authors introduce the external focus (EF) of attention in relation to implicit learning.

While there are indeed indications that an EF indeed leads to a more implicit learning

process, the majority of the studies on EF (at least when compared to an internal focus) use a

different theoretical framework (constrained action hypothesis). I would suggest that the

authors include a more elaborate theoretical framework, acknowledging the different ‘lines

of research’ on this topic which can strengthen their argumentation for the specific research

questions and hypotheses. 

Reply: thank you for your suggestion. 

Implicit and explicit learning can be shaped in various ways. Relevant implicit strategies are: (1) EF learning; (2) analogy learning; (3) errorless learning; and (4) dual task learning. The constrained action hypothesis is a theoretical framework underlying one of these strategies, namely the focus of attention. 

We mentioned in the manuscript (p4, lines 58-59) that “Implicit learning can be shaped by using an external focus of attention (EF) (23).” This now reads: “Implicit learning can, for instance, be shaped by using an external focus of attention (EF) (23).”

Furthermore, we added the constrained action hypothesis and two new references in the introduction (references 25 and 26) (p4, lines 58-64):

• A new reference for the description of EF and IF, because this one felt better suited.

• A new reference supporting the constrained action hypothesis. 

This now reads: “Implicit motor learning can, for instance, be shaped by using an external focus of attention (EF) (23). With an EF, the child’s attention is directed to the impact of the movement on the environment (25). On the contrary, with an internal focus of attention (IF) the attention is directed to its body movements (25). According to the constrained action hypothesis, an IF promotes a larger involvement of cognitive processes due to a greater reliance on conscious control strategies. These strategies interfere with the normal automatic control processes of the motor system. An EF promotes these automatic control processes, therefore, enhancing motor learning more (26).”

2. In addition to this point I would suggest that the authors briefly explain why they only focus

on an EF and on aspects that can be controlled by an instructor (frequency, timing and form).

For example, the authors also mention that an IF might be more beneficial is specific

populations (line 65), and mention the moderating effect of child and task characteristics

(line 78). 

Reply: we acknowledge that there are several knowledge gaps about the use of instructions and feedback in children’s motor learning:

1. In scientific research, most studies focus on one single aspect (so-called parameter) of instructions and feedback. For instance, they investigate either the focus, frequency, timing or form. However, in clinical practice, it is the combination of these parameters that shape instructors’ instructions and feedback (p4, lines 73-74). Yet, little is known about the effectiveness of instructions and feedback in which parameters were combined (p5, lines 90-91). 

2. Indeed, there are some studies that showed that an IF is more effective than an EF suggesting that effectiveness is moderated by child characteristics. Two recent systematic reviews in children’s motor learning also suggested that effectiveness is moderated by child and task characteristics (van Abswoude et al., 2021; Simpson et al., 2020). However, more research is necessary to gain insights into which child and task characteristics are relevant, and how they moderate effectiveness.

For this systematic review, we decided to focus on the effectiveness of instructions and feedback in which at least two parameters were combined, because it gained insights into how instructors can shape their instructions and feedback. A next step would be to investigate the moderating effect of child and task characteristics. However, we believe that more studies of methodological good quality are needed first. For further explanation we refer to the reply on comment 5.

We focused an EF because of its suggested beneficial effects according to the constrained action hypothesis. Therefore, we added a sentence to argue our choice (p4, lines 71-72). 

This now reads: “Although, the beneficial effects of the EF have not yet been shown for each population, the constrained action hypothesis promotes using an EF for teaching motor skills (26). Therefore, this systematic review focuses instructions and feedback with EF.”

3. A final comment regarding the introduction would be to strengthen the argumentation for

the hypotheses that are given at the start of the method section. In the paragraph starting at

line 68 a very brief explanation of the specific ‘instructor controlled’ aspect of EF feedback,

but I would prefer to know more about the how and why. For example, in line 73-74 the

authors mention that ‘Self-controlled feedback may enhance children’s motor learning more

than instructor-controlled feedback’. Why is this the case, what is the mechanism? And is this

specifically related to an EF. 

Reply: the theoretical framework underlying our timing hypothesis is the Self-Determination Theory, which is not specifically related to the EF. We added additional information to the original text (p5, lines 85-87). This now reads: “Self-controlled timing advances a child’s autonomy, which is essential to enhance intrinsic motivation according to the Self-Determination Theory (37). As motivation is considered relevant in motor learning, self-controlled timing could be more effective (38). Studies in children showed that self-controlled feedback may enhance motor learning more than instructor-controlled feedback (36).”

The theoretical framework underlying our frequency hypothesis is the guidance hypothesis. We added additional information to the original text (p4, lines 76-79). This now reads: “Based on the guidance hypothesis, a reduced frequency would be more beneficial for retention and transfer than a continuous frequency because it reduces the feedback dependency enhancing the processing of other sources of information, which results in more implicit learning (34).”

There is no specific theoretical framework underlying the form hypothesis. In clinical settings, demonstrations are often used to teach children motor skills which seems effective for all types of children, typically and atypically developing. However, most scientific studies used verbal instructions and feedback, therefore, we focused on other forms of instructions and feedback.

Methods

4. Given the different theoretical backgrounds that studies on EF may have, I would suggest to

include a clear definition as to what the authors include as an EF. For example, in the

included studies I saw many that use knowledge of results as a form of feedback, which may

not be regarded as an EF when compared to the literature that use the constrained action

hypothesis as a theoretical framework. 

Reply: we acknowledge that EF and KR are different. 

The article of Wulf et al. (2001) introduces the constrained action hypothesis. Following is written in the introduction: “In two experiments, Wulf et al. (1998) demonstrated the greater effectiveness of instructions that induced an external focus of attention (i.e., directed the performer's attention to the movement effect) as compared to those that induced an internal focus of attention (i.e., directed attention to the movements themselves).” 

In the review of Salmoni et al. (1984) KR is described as: “information provided after a response that tells of the learner’s success in meeting the environmental goal”. The article of Winstein et al. (1990) described KR as: “KR refers to the extrinsic information about task success provided to the performer after a practice trail has been completed. It is considered a subset of feedback, which is augmented, verbal (or verbalizable), post response information about the movement outcome in terms of the environmental goal. This information serves as a basis for error corrections of the next trial and as such can lead to more effective performance as practice continues”.

Although EF and KR are described differently, both provide the child with information about the results of the movement on the environment. In EF-instructions or feedback, the child is told by the instructor how to act, whether in KR, the instructor informs the child about the results and the child needs to process this information to determine how to act. Therefore, we considered KR as a subtype of EF.

We added extra information to inclusion criteria 2 providing insights into the differences and why we included both (p6, lines 117-125). 

This now reads: “With instructions or feedback with EF the instructor directs the attention of the child to the effects of the movement on the environment (e.g. “Try to focus on the red markers and try to keep the markers at the same height” when balancing a stabilometer) (25). With Knowledge of Results feedback (KR) the instructor informs the child about the effects of the movement on the environment (e.g. by indicating to what extent the ball deviated the target in direction and distance) (41). This information serves as a basis for error corrections improving next performances (34). Although in KR the child needs to process the obtained information more to determine how to act, both EF and KR focus on the effects of the movement on the environment. Therefore, we considered KR as a subtype of feedback with EF.”

5. I agree with the authors that a meta-analysis on the data is not possible and their decision to

perform a best-evidence synthesis. I do, however, want to question the decision to combine

the results of typically developing and a-typically developing children in their rating of the

evidence for a parameter of interest. The authors may want to revisit this decision, or at least

provide a rationale and discuss the implications in the discussion, as this greatly influences

the interpretation of the findings and the conclusions. 

Reply: thank you for your comment. We have considered your suggestion carefully; however, we finally have decided not to perform sub analyses in typical and atypical populations. We will underpin our reasons below.

We aimed to investigate whether instructor-controlled factors like frequency, timing and form influenced effectiveness of instructions and feedback in children. A first step, in line with our chosen in and exclusion criteria, is to perform analyses with all children combined.

A subsequent step could be to perform sub analyses. However, on which relevant variables are we going to make our choice(s)? Variables are relevant if we expect that there will be a relationship between selected instructor-controlled factors and the effectiveness of instructions and feedback in children. It can be assumed that effectiveness would differ between typical and atypical population due to differences in cognitive functioning (which influences the processing of the received instructions and feedback). However, other child characteristics like age, motor abilities and motivational factors are also likely to influence effectiveness as well (Simpson et al., 2020). In order to improve insights into the moderating effect of child characteristics, sub analyses could be performed on all relevant characteristics. However, given the sample of included studies, and additional knowledge in the literature regarding potentially relevant variables, we have made the choice in this manuscript not to do so. Our reasons are summarized below: 1. insufficient insights, and presented data in the included studies, into which characteristics could be potentially relevant; and 2. the limited number of studies and, foremost, the low methodological quality of the studies. Therefore, we decided not to perform sub analyses. However, we acknowledge that it is an important subsequent step including all potentially relevant variables that should have attention in future research when more studies of methodological sound quality are available.

We made following changes in the manuscript to elaborate on our choice and to discuss implications:

• Methods – section “analyses” (p11, lines 226-231): we added the argument for not performing sub analyses. This now reads: “This study aimed to investigate whether the instructor-controlled parameters frequency, timing and form moderate effectiveness of instructions and feedback in children. Subsequent analyses with sub groups were not performed for two reasons: 1. it was not possible to define relevant sub groups due to insufficient insights, and presented data in the included studies, into which child characteristics could be potentially relevant to moderate effectiveness (36); and 2. the number of studies per potential comparison and methodological quality was too low.”

• Discussion (p21, lines 490-497): we discussed implications acknowledging the relevance of adapting instructions and feedback to child and task (with two references, references 17 and 105). This now reads: “Secondly, it is suggested that child and task characteristic may moderate effectiveness (23,36). However, more research is necessary to gain insights into which characteristics are relevant, and their moderating role. Accordingly, it was not possible to perform sub analyses in the best-evidence synthesis. For instructors, it is not only important to know how to shape their instructions and feedback, but also how to adapt their instructions and feedback to child and task (17,105). Therefore, performing sub analyses on all potentially relevant variables such as typical/atypical development, age, cognitive or motivational factors, would be recommended for future research when more methodologically sound studies are available, including relevant data to make sub groups properly.”

Following adaptation was made as well:

• Table 2: The task and population were mentioned in one column. To provide the reader with the opportunity to better compare the data per population, we splitted this column into two separate columns, and re-ordered the studies by population. 

Results

6. Did you, or the literature that you are reviewing, consider to also include if groups improve

their performance over time instead of only focusing on group differences at a specific time

point? I would argue that if children do not improve their performance (which may be the

case with some very short practice periods) it is not surprising that group differences are also

lacking. Therefore, this lack of a significant effect may not be a good representation of the

effect that you are most interested in (specific feedback parameters) 

Reply: we agree with your arguments that it is relevant information to report. However, within an RCT the primary goal is the comparison between groups and should be leading in reporting and determining success of a trial. The within comparison is secondary in a trial. Although the within comparison is very important in clinical practice and for the individual child, it could not be interpreted as an effect of the given intervention because other (not investigated) variables could have potential influence on the change scores.

Twelve out of 13 studies showed within group improvements during practice. We added this information in the manuscript under results – “study characteristics” (p13, lines 284-285)

This now reads: “All groups showed within group improvements during practice in 12 out of 13 studies (41,78,87,90,79–86)”

Furthermore, in table 1 (column “results”), we provided additional information about whether groups showed within group improvements during acquisition.

These now reads:

Example 1: de Oliveira et al., 2009

ANOVA analysis with post-hoc Tukey HSD tests 

Accuracy

· During acquisition:

Significant main effect for blocks, accuracy improved during practice

Significant main effect for frequency

Post-hoc testing: 25% > 50%, 75% and 100%* and 75% > 50%*

Example 2: Hemayatallab & Rostami, 2010

Significant paired sample t-tests for pre-test / post-test for all groups, all groups improved accuracy

ANOVA analysis with post-hoc Tukey HSD tests

· Post-test: 

Significant main effect for frequency 

Post-hoc testing: 100% > 50%**, 100% > 0%** and 50% > 0%*

About table 1, we added the information that the studies using two outcome measures (de Olivera et al., 2009; Sidaway et al., 2012; Wulf et al., 2010) did not report which outcome was the primary and secondary outcome measure. We added this information by using the c with explanation in the legend.

Discussion

7. As also mentioned in comment 5, I question the interpretation of the findings when they are

based on a combination studies on both typically- and a-typically developing children. For

example, the majority of the studies (5 total) found a beneficial effect on retention for

reduced frequency, and only one study showed better retention with continuous frequency,

but in children with ASD, leading to conflicting evidence. On the other hand, limited evidence

for continuous frequency, as opposed to reduced frequency, is claimed with only one study

performed for movement quality. Also, the suggestion that visual feedback might be

preferred is based on 1 study in children with MID and ASD. I believe the discussion should

be better balanced by acknowledging how characteristics of theses specific groups may have

influenced the outcomes of the individual studies and with that the overall interpretation of

the evidence. 

Reply: we reorganized and expanded the paragraphs in the discussion of the findings of the frequency (p18) and timing (p19). We elaborated on the individual studies and the characteristics of the populations. Due to the limited number of studies, it occurred that conclusions had to be drawn on one study of low methodological quality, underpinning the need for more studies in this domain. However, we argued why we did not perform sub analyses and acknowledged the importance of future research to gain more insights into the moderating role of child characteristics (see comment 5).

For the paragraph about frequency we made following changes:

• The limited evidence supporting the continuous frequency to improve quality of movement appeared incorrect and is removed. (See minor comment 6)

• The limited evidence supporting the continuous frequency in comparison to the faded frequency is replaced to the end of the paragraph, giving more accent on the discussion about the differences between studies (next bullet point). Also, we added that the study had a one-day training protocol. This now reads (p18, lines 393-395): “Only one study compared a faded reduced frequency to a continuous frequency in TDC using a one-day training protocol, resulting in limited evidence for continuous frequency for retention (78).”

• We focused more on the differences between studies; added that the conflicting results resulted from studies favoring the reduced frequency (as expected) and studies with non-significant results. The low methodological quality of the studies made it difficult to formulate conclusions about significance of results. This now reads (p18, lines 389-396): ”For retention, conflicting evidence was found for accuracy as well, however, four out of seven experiments found beneficial effects for reduced frequency (83,84,88,90) as expected (34). From the remaining three studies, two found non-significant results (81,82). For transfer, no evidence was found for accuracy (81,82,85). However, these studies, also measuring variability and quality of movement, found non-significant results for acquisition and retention as well (81,82,85). Only one study compared a faded reduced frequency to a continuous frequency in TDC using a one-day training protocol, resulting in limited evidence for continuous frequency for retention (78). The interpretation of these results might be influenced due to methodological limitations, which will be elaborated later.”

• Recently, a new systematic review on the frequency of feedback in typically developing children and children with Cerebral Palsy was published. We added this review to the reference list (reference 33), and rewrote the evidence of existing literature. This now reads (p18/19, lines 397-415): “Systematic reviews investigating effectiveness of frequency of feedback to improve motor skills in TDC and children with CP found limited or contradicting evidence for children with CP (32,33). They suggested that child characteristics and task complexity might moderate effectiveness, but foremost they recommended that more studies of methodologically sound quality are needed to draw conclusions (32,33). For TDC, they concluded that reduced frequency might be more effective (33). However, two studies investigating the effectiveness of reduced frequency in TDC and CP did not include a control group with continuous frequency. Furthermore, the study that compared continuous with faded frequency found no differences between groups for TDC (33).”

• Based on the rewriting of bullet point 3, we also rewrote the final sentence to close this paragraph. This now reads (p19, lines 415-417): “In summary, several individual studies in the best-evidence synthesis showed beneficial effects for reduced frequency for retention, and for continuous frequency for acquisition. However, overall results in this, and previous studies, were conflicting.”

For the paragraph about timing we made following changes:

• In line with our argument in the frequency-paragraph that the duration of the training protocol might explain why continuous feedback appeared more effective, we added the one-day training protocol to following sentence (p19, lines 423-424). This now reads: “This inconsistency may be due to the frequency of feedback, as the self-controlled group received less feedback than the instructor-controlled group during the one-day training protocol (78).”

• We elaborated the results of the individual studies (p19, lines 424-427). This now reads: “For all other time points, either no or conflicting evidence was found. However, if results were conflicting, studies found either non-significant results or evidence favouring self-controlled timing as was expected by the Self-Determination Theory (41,80,86,87).”

• We elaborated on the role of cognitive deficits in self-controlled timing related to the atypical population. This now reads: “In the best-evidence synthesis, three out of four studies with equal frequency of feedback in both groups included children with either CP (86,87) or DCD (80). These children are characterized by cognitive deficits, which might influence their abilities for autonomous functioning (6,37,94). These characteristics, in addition to the methodological limitations, might explain why results are not as consistent as expected (37).”

• We deleted the text that “The advantages of self-controlled feedback were also found in adults” because with the expansions this became less relevant.

8. Overall, the discussion had a large focus on the practical value of the outcomes, but is

missing the more theoretical explanations and implications for the (inconsistent) results,

which links to my comments about the theoretical framework in the introduction (comment

1 and 3). 

Reply: we added more links to theoretical frameworks and elaborated the discussion.

• We added the constrained action hypothesis to the introduction paragraph (P17, lines 373-374): “Although, the constrained action hypothesis suggested that an EF would be more effective, previous research investigating effectiveness of instructions or feedback with EF found conflicting results for children (23,36) and adults (43,91). It was hypothesized that the frequency, timing and/or form of instructions and feedback (20) influenced their effectiveness.”

• For the frequency-paragraph:

o We rewrote the sentence about the amount of information needed in early learning stages to link it to the guiding hypothesis (p18, lines 383-385). This now reads: “At the beginning of the learning process, feedback dependency is likely to be higher because more information (e.g. by means of more instructions and feedback) is needed to acquire new skills (12,92,93).”

o We elaborated on the differences between the individual studies (see also reply comment 7) mentioning more explicitly that several individual studies found beneficial effects for reduced frequency as expected.

• For the timing-paragraph: we had already mentioned that cognitive skills might influence effectiveness of self-controlled feedback. We added that several studies included atypical populations with cognitive deficits which might influence the abilities of autonomous functioning (see also reply comment 7). 

9. In lines 343-350 the authors describe the outcomes of a previous systematic review on the

frequency of feedback on motor learning. Given the arguments presented here, I would

question if the hypothesis that the authors have stated regarding the benefits of reduced

feedback is correct? 

Reply: we based our hypothesis for frequency on the guiding hypothesis (Winstein et al. 1990), which we added as theoretical framework to the introduction (p4, lines 76-79). This now reads: “Based on the guidance hypothesis, a reduced frequency would be more beneficial for retention and transfer than a continuous frequency because it reduces the feedback dependency enhancing the processing of other sources of information, which results in more implicit learning (34).”

The review (Roberts et al., 2017) mentioned in lines 343-350 found limited evidence (Sackett’s level 2b, based on two studies) that continuous frequency would be more beneficial for TDC for acquisition, and reduced faded frequency for retention. Both studies used upper limb laboratory tasks, which are different types of tasks than the functional tasks we included. For reduced compared to continuous frequency they found conflicting results (Sackett’s level 2b, based on four studies with throwing task). For CP the evidence was very limited (three pre-post design studies, Sackett’s level 3) showing that both faded and continuous frequency appeared effective. The presented evidence as was written in the manuscript might suggest that our hypothesis should have been differently, but all evidence is limited due to methodological low quality of the studies. Therefore, we supported our hypothesis with a theoretical framework.

Recently another systematic review (Schoenmaker et al., 2022) on feedback frequency in TDC and CP was published. We added this review to the reference list (reference 33) and rewrote the text about the findings of previous studies (see reply comment 7). They described the results of the individual studies without providing an overall level of evidence. The individual studies concerning children with CP showed contradicting results. For TDC, they concluded that reduced frequency might be more effective (based on two studies with no control group and one study with a control group that did not found differences between groups). Both studies concluded that more studies of methodologically sound quality are needed to draw conclusions about the effectiveness of different frequencies of feedback. We addressed this message more explicitly (p18, lines 399-401). 

This now reads: “They suggested that child characteristics and task complexity might moderate effectiveness, but foremost they recommended that more studies of methodologically sound quality including the investigation of relevant child characteristics are needed to draw conclusions (32,33).”

10. The suggestion made in lines 362-365 should be placed a bit more cautiously. That is, while

there are some benefits shown for self-controlled feedback, 2 studies compare this with a

yoked condition. It can be argued that the moments when an instructor provides feedback,

are more helpful than the moments an experimental counterpart chose this. Also, according

to table 1 the study of Hemayattalab (2014) did not show a difference between self-

controlled and instructor-controlled feedback, which contradicts your suggestion.

Reply: because the focus of our interest is the self-controlled timing, and the number of studies were limited, we combined yoked and instructor-controlled groups as control. We added this explicitly to the method, section analyses (p10, lines 210-214). 

This now reads: “In studies investigating timing, the control group is either yoked (the children received feedback as their counterpart in the intervention group requested feedback) or instructor-controlled (the instructor determined when the child received feedback). Because of the chosen focus of this systematic review in the self-controlled aspect, we combined both yoked and instructor-controlled groups as control intervention.” 

However, we explored your argument that the timing of the feedback differs between instructor-controlled (IC) and yoked control conditions. For the reason that the timing of the instructor should be more beneficial (because the instructor would choose a more helpful moment) than in the yoked group, you would expect that the differences between the self-controlled (SC) group and IC group would be smaller (because both child and instructor choose the most helpful moment) resulting in higher chance to find non-significant results. On the contrary, you would expect that the differences between groups would be larger in the studies in which the SC group was compared to the yoked group (because the SC would choose the most helpful moment, while their counterparts in the yoked group did not have this opportunity) resulting in higher chance to find significant differences. As seen in following table, the assumption holds for retention and transfer. However, the number of studies was limited and the methodological quality was low. Therefore, also in line with our chosen focus, we decided not to perform sub analyses per control condition, but to elaborate on it in the discussion. 

This now reads (p19, lines 428-434): ”In this study, the yoked and instructor-controlled groups were combined as control. However, it can be argued that effectiveness can differ depending on the type of control group. Moreover, instructor-controlled feedback may be more supportive to the child than the yoked controlled feedback because of its timing; it is to be expected that the instructor estimates when the feedback would be most informative to the child, while in the yoked condition the moment of feedback is not related to the child’s performances. It would be interesting to explore this assumption in future research.”

 

 Timing Self-controlled vs yoked

(equal frequency in both groups) Chiviacowsky

et al. 2008 TDC Throw with beanbag NS NA 24h SC NA NA - NA **

SC *

SC

 Hemayattalab

et al. 2013 CP Throw with beanbag NS NA 24h SC 24h SC 

 Self-controlled vs instructor-controlled (equal frequency in both groups) Zamani et al. 2015 DCD and MID Throw with tennis ball NA NS NR SC NA NA *

SC - X -

 Hemayattalab

et al. 2014 CP Dart throwing SC NA 24h NS 24h NS 

Furthermore, we are aware that the study of Hemayatallab et al. (2014) showed different results. However, in the summary synthesis 75% of the studies showed results in the same directions, which is considered consistent, resulting in moderate evidence. 

11. In line 403-404 the authors mention that instructors should adapt instructions to the

individual (among other things). I agree with these suggestions and given the different

populations included in this review (and their specific characteristics) I wondered if the

authors could give some informed suggestions about which aspects of the individual

instructors may want to focus on

Reply: an informed suggestion is difficult, because more research of methodologically sound quality is necessary to gain a better understanding of which child characteristics are (most) relevant. However, we suggested some relevant characteristics based on previous research (p22, lines 505-510).

This now reads: “This framework, and other studies, suggest that instructors should adapt frequency, timing and form of instructions and feedback to the individual and the task (17,23,36,92,105). Child characteristics as skill level, cognitive functioning, motivation, and the presence of a diagnose are considered relevant (17,23,36,105). However, more research is necessary to gain a better understanding of their moderating role. Therefore, future research should attempt to include a wider variety of tasks and/or child characteristics in their studies.

 

Minor issues/comments

Methods

1. Line 122. Tactile instructions are mentioned as an exclusion criterion for the control condition

given the IF, but are not all instructions that may promote an IF excluded? 

Reply: Indeed, all instructions and feedback with IF were excluded, which was not mentioned explicitly in the methods of the manuscript. We added this information to exclusion criterion 2 (intervention) (p7, line 141). This now reads: “2. Intervention: Instructions or feedback with an IF; intervention methods like Neuromotor Task Training, because they provide no insight into effectiveness of separate instructions or feedback; instructions and feedback used to learn laboratory, fine motor and static balance tasks, because they did not meet the definition of functional gross motor task (2).”

For tactile form, we rewrote the sentence making more explicitly why a tactile focus enhances an IF (p7, lines 145-146). This now reads: “3. Control: A tactile form of instructions and feedback, because it directs the attention of the child to the body, therefore, promoting an IF.”

2. I would like some numbers or percentages on the (dis)agreement between authors regarding

study selection and methodological quality assessment

Reply: we added the percentages of (dis)agreements.

For selection, this now reads (p12, lines 244-247): “The search resulted in 3813 unique hits. After screening title and abstract, 3521 hits were excluded. The reviewers agreed in 86% of the studies on inclusion or exclusion, 14% of the abstracts were discussed. The remaining 292 hits were screened on full text, eight of which met the inclusion criteria. The reviewers agreed in 93% of the studies on inclusion or exclusion, 7% of the articles were discussed.”

Although, the documents for risk of bias assessment were discussed between reviewers in preparation of the risk of bias assessment, percentages of agreement varied. Therefore, an epidemiologist was present during the consensus meeting to provide additional insights. 

We added extra information to the methods – section “methodological quality assessment”. This now reads (p10, lines 202-203). This now reads: “A consensus meeting was organized with all reviewers and an epidemiologist (CB) to reach consensus.”

Furthermore, we added the percentages of agreements to the results – section “methodological quality” (p12, lines 260-261). This now reads: “Percentages of agreements between reviewers varied (Domain 1 75%; Domain 2 25%; Domain 3 41%; Domain 4 25%; Domain 5 67%).” And (p13, lines 272): “Reviewers scored similar for all domains except Domain 6.”

In the discussion we changed “third reviewer” to “epidemiologist” (p20, line 465). This now reads: “A strength of this study was that it followed a registered protocol, comprising a selection process and RoB assessment performed by two reviewers independently, with an epidemiologist (CB) to be consulted in cases of disagreement.”

Results

3. In table 1 I would suggest to include more information about the specific feedback or

instructions that were given, as this is one of the main characteristics of interest in this

review. 

Reply: thank you for this suggestion, we added a description of the used instructions and feedback to the column “groups” in table 1.

These now reads:

Example 1: Gillespie et al., 2003

Groups

· 20% frequency: KR after every fifth triala

· 100% frequency: KR after every triala

The provided feedback 

the instructor informed the child about the accuracy score and the child was allowed to see where the ball had stopped

Example 2: Tse & Masters, 2019

Groups

· Instruction with visual analogy (n = 12)

· Instruction with verbal analogy (n = 12)

· Explicit instruction: with IF on arm or hand (n = 12)

· Control: no specific instruction (n=12)

The provided instructions

• Visual analogy: an illustration of a child putting a cookie in a cookie jar on a high shelf

• Verbal analogy: “Shoot the ball as if you are trying to put cookies into a cookie jar on a high shelf”

• Explicit instruction: “Move the ball upward and release the ball when your strong arm becomes vertical”; “When releasing the ball, your strong hand is facing downwards”

4. I would like to suggest some re-ordering in the results section. I would prefer the description

of the methodological quality a bit earlier in the results. Also, you may want to consider

combining the general description of studies focused on a specific parameter (paragraphs

starting at lines 230, 239 and 247) with the best-evidence synthesis of that parameter. I was

now going back and forth between these sub-sections for a complete overview (and you may

also prevent writing info in 2 places). 

Reply: Thank you for this suggestion, we re-ordered the results and switched sections “methodological quality” and “study characteristics”. We did not merge the section “study characteristics” and “best-evidence synthesis”. By reordering sections, the information will follow now in more logical order which already may reduce the reader to go back and forth within the text.

5. Line 274, the order of the references seems mixed (72, 73, 82-84, 74, 81)? 

Reply: we checked every reference order, and corrected it if necessary. 

6. There is an inconsistency between table 1 and 2 regarding the study of Wulf (2010), where

table 1 shows no difference between groups, and table 2 does show a difference for

movement quality. 

Reply: thank you very much for noticing this error. The information in table 1 was correct. We changed table 2 (last row, last two columns), and corrected it in the manuscript within the results (p16, lines 340-342). 

This now reads: “There was no evidence that reduced fixed or continuous frequency was more effective in reducing variability or improving quality of movement in throwing in TDC for acquisition, retention and transfer”. 

Furthermore, we removed following sentence in the discussion (p18): “Also, limited evidence favoured continuous frequency to improve quality of movement in soccer ball throwing in TDC for retention and transfer.” And corrected it in the conclusion as well (p22).

7. Line 317, the authors mention 33% self-controlled feedback, whereas table 1 shows that the feedback is for 3 out of 10 trials, which should be 30%. 

Reply: it should indeed be 30%. We corrected it in table 2, and in the manuscript (p14, line 300; p17, line 364)

Discussion

8. Given the practical value that the authors stress in the discussion, they may want to mention

aspects of ecological validity of the studies included in the review in the discussion

Reply: thank you for this suggestion. Although, we did not use the term ecological validity explicitly, we did mention several aspects of it in the discussion. We rewrote some sentences to underline the ecological validity.

• It was added as a strength of this study (p20, line 469). This now reads: “This study included functional tasks which improved the ecological validity of this study.”

• It was added to the recommendation we did about adopting longer practice duration in future research (p18, lines 387-389). This now reads: “Therefore, future studies adopting longer practice durations would be of more practical interest which will improve ecological validity as well.”

• It was added to the recommendation we did about adopting more type of tasks and child characteristics in future studies (p22, lines 509-511). This now reads: “Therefore, future research should attempt to include a wider variety of tasks and/or child characteristics in their studies. This will improve ecological validity and generalizability of the studies as well.”

• It was added to the recommendation we did to use more types of outcomes in future studies (p22/23, lines 519-527, it skips 5 lines in numbering). This now reads: “Therefore, for better ecological validity, more result-related outcomes (e.g. variability, number of successful attempts and distance) and movement pattern-related outcomes (e.g. quality of movement and kinematic variables) should be considered in future studies.”

• Furthermore, we expanded our recommendation for future research by mentioning all variables to improve ecological validity in future studies (p23, lines 536-539). This now reads: Furthermore, it should aim to advance insights into the modifying role of frequency, timing and form in instructions and feedback with EF with methodologically sound studies focusing on: 1. a variety of tasks; 2. populations with different skill levels, age ranges, and diagnoses; 3. various outcome measures; and 4. with longer practice duration.”

Reviewer #2: 

This systematic review is focused on the role played by frequency, timing and form on the performance of functional skills. The authors reported some possible evidence for using self-controlled feedback, visual instructions and a continuous frequency of feedback. The review deals with an interesting topic and has some potentialities. I have only some concerning about the Authors should consider.

The introduction is focused and well-written, the scope is clear, and the bibliography updated.

The methods are well-structed and the procedure used are accurate. I have only a concern about the merging results from children with a typical and an atypical development. This should be considered and discussed.

The results are well reported

The discussion is well supported by the data provided by the meta-analysis. As above stated, I would reconsider the part coming from the combination of study including typically and atypically developing children.

Reply: thank you for your positive comments and constructive feedback. We have considered your suggestion carefully; however, we finally have decided not to perform sub analyses in typical and atypical populations. We will underpin our reasons below.

We aimed to investigate whether instructor-controlled factors like frequency, timing and form influenced effectiveness of instructions and feedback in children. A first step, in line with our chosen in and exclusion criteria, is to perform analyses with all children combined.

A subsequent step could be to perform sub analyses. However, on which relevant variables are we going to make our choice(s)? Variables are relevant if we expect that there will be a relationship between selected instructor-controlled factors and the effectiveness of instructions and feedback in children. It can be assumed that effectiveness would differ between typical and atypical population due to differences in cognitive functioning (which influences the processing of the received instructions and feedback). However, other child characteristics like age, motor abilities and motivational factors are also likely to influence effectiveness as well (Simpson et al., 2020). In order to improve insights into the moderating effect of child characteristics, sub analyses could be performed on all relevant characteristics. However, given the sample of included studies, and additional knowledge in the literature regarding potentially relevant variables, we have made the choice in this manuscript not to do so. Our reasons are summarized below: 1. insufficient insights, and presented data in the included studies, into which characteristics could be potentially relevant; and 2. the limited number of studies and, foremost, the low methodological quality of the studies. Therefore, we decided not to perform sub analyses. However, we acknowledge that it is an important subsequent step including all potentially relevant variables that should have attention in future research when more studies of methodological sound quality are available.

We made following changes in the manuscript to elaborate on our choice and to discuss implications:

• Methods – section “analyses” (p11, lines 226-231): we added the argument for not performing sub analyses. This now reads: “This study aimed to investigate whether the instructor-controlled parameters frequency, timing and form moderate effectiveness of instructions and feedback in children. Subsequent analyses with sub groups were not performed for two reasons: 1. it was not possible to define relevant sub groups due to insufficient insights, and presented data in the included studies, into which child characteristics could be potentially relevant to moderate effectiveness (36); and 2. the number of studies per potential comparison and methodological quality was too low.”

• Discussion (p21, lines 490-497): we discussed implications acknowledging the relevance of adapting instructions and feedback to child and task (with two references, references 17 and 105). This now reads: “Secondly, it is suggested that child and task characteristic may moderate effectiveness (23,36). However, more research is necessary to gain insights into which characteristics are relevant, and their moderating role. Accordingly, it was not possible to perform sub analyses in the best-evidence synthesis. For instructors, it is not only important to know how to shape their instructions and feedback, but also how to adapt their instructions and feedback to child and task (17,105). Therefore, performing sub analyses on all potentially relevant variables such as typical/atypical development, age, cognitive or motivational factors, would be recommended for future research when more methodologically sound studies are available, including relevant data to make sub groups properly.”

---

## [Decision Letter · Decision Letter 1]

8 Aug 2022

How can instructions and feedback with external focus be shaped to enhance motor learning in children? A systematic review

PONE-D-22-04840R1

Dear Dr. van der Veer,

We’re pleased to inform you that your manuscript has been judged scientifically suitable for publication and will be formally accepted for publication once it meets all outstanding technical requirements.

Kind regards,

Emiliano Cè

Academic Editor

PLOS ONE

Additional Editor Comments (optional):

Reviewers' comments:

Reviewer's Responses to Questions

**Comments to the Author**

1. If the authors have adequately addressed your comments raised in a previous round of review and you feel that this manuscript is now acceptable for publication, you may indicate that here to bypass the “Comments to the Author” section, enter your conflict of interest statement in the “Confidential to Editor” section, and submit your "Accept" recommendation.

Reviewer #2: (No Response)

2. Is the manuscript technically sound, and do the data support the conclusions?

Reviewer #2: Yes

3. Has the statistical analysis been performed appropriately and rigorously? 

Reviewer #2: N/A

4. Have the authors made all data underlying the findings in their manuscript fully available?

Reviewer #2: (No Response)

5. Is the manuscript presented in an intelligible fashion and written in standard English?

Reviewer #2: Yes

6. Review Comments to the Author

Reviewer #2: The Authors have responded to all my previous comments. I therefore recommend the acceptance of the manuscript as in the present form.

7. PLOS authors have the option to publish the peer review history of their article (what does this mean?). If published, this will include your full peer review and any attached files.

Reviewer #2: No

---

## [Editor Report · Acceptance letter]

16 Aug 2022

PONE-D-22-04840R1 

How can instructions and feedback with external focus be shaped to enhance motor learning in children? A systematic review 

Dear Dr. van der Veer:

I'm pleased to inform you that your manuscript has been deemed suitable for publication in PLOS ONE. Congratulations! Your manuscript is now with our production department. 

Kind regards, 

on behalf of

Professor Emiliano Cè 

Academic Editor

PLOS ONE